# GROUNDING MULTIMODAL LLMS TO EMBODIED AGENTS THAT ASK FOR HELP WITH REINFORCEMENT LEARNING

**Ram Ramrakhya**[1]* **Matthew Chang**[2] **Xavier Puig**[2] **Ruta Desai**[2]
**Zsolt Kira**[1] **Roozbeh Mottaghi**[2]
[1]Georgia Institute of Technology [2]Meta FAIR

## ABSTRACT

Embodied agents operating in household environments must interpret ambiguous and under-specified human instructions. A capable household robot should recognize ambiguity and ask relevant clarification questions to infer the user intent accurately, leading to more effective task execution. To study this problem, we introduce the ASK-TO-ACT task, where an embodied agent is tasked with a single or multi-object rearrangement task using an under-specified instruction in a home environment. The agent must strategically ask minimal, yet relevant, clarification questions to resolve ambiguity while navigating under partial observability. To address this challenge, we propose a novel approach that fine-tunes multi-modal large language models (MLLMs) as vision-language-action (VLA) policies using online reinforcement learning (RL) with LLM-generated rewards. Our method eliminates the need for large-scale human demonstrations or manually engineered rewards for training such agents. We benchmark against strong zero-shot baselines including GPT-4o as well as supervised fine-tuned MLLMs on our task. Our results show that our RL-finetuned MLLM outperforms all baselines by a significant margin (10.4-16.5%), generalizing well to novel scenes and tasks. To the best of our knowledge, this is the first demonstration of adapting MLLMs as VLA agents that can act and ask for help using LLM-generated rewards with online RL.

## 1 INTRODUCTION

Embodied agents interacting with humans must interpret human instructions that are often ambiguous, under-specified, or context-dependent. Consider asking a household robot, '*Bring the cup and place it on the coffee table*' To complete this task in an environment (as shown in Fig. 1) without additional information, the robot faces multiple ambiguities like - which *cup* is the user referring to (red, white or green)? Does the user need a small cup or a large one? Instead of making incorrect assumptions and bringing an incorrect cup to the user, a capable household robot should ask the *minimum number of* clarification questions to disambiguate the requested object such as, *"Are you looking for a red cup?"* followed by *"Is it on the kitchen counter?"*. By actively asking questions that are grounded in the environment state and relevant to the context of the task, an intelligent household robot can infer user intent and preferences more accurately, leading to more effective and reliable task execution. We call this task ASK-TO-ACT, and believe inferring such ambiguity or user preference and asking clarification questions to resolve it requires balancing several implicit objectives such as exploration to gather task-specific information from the environment, deductive reasoning to minimize the number of questions, and contextual reasoning to determine the right time to ask questions. Humans effortlessly do this by combining common sense knowledge, contextual cues, and deductive reasoning. In this work, we ask 'How can we train embodied agents capable of reasoning required to resolve ambiguity and user preferences by asking clarification questions?'

Training such agents capable of interacting and asking clarification questions is challenging. Most tasks in embodied AI have well-defined goals (Batra et al., 2020; Weihs et al., 2021; Ehsani et al., 2021; Szot et al., 2021) and approaches for such tasks rely on either collecting large-scale human

---
*Work done as part of internship at Meta

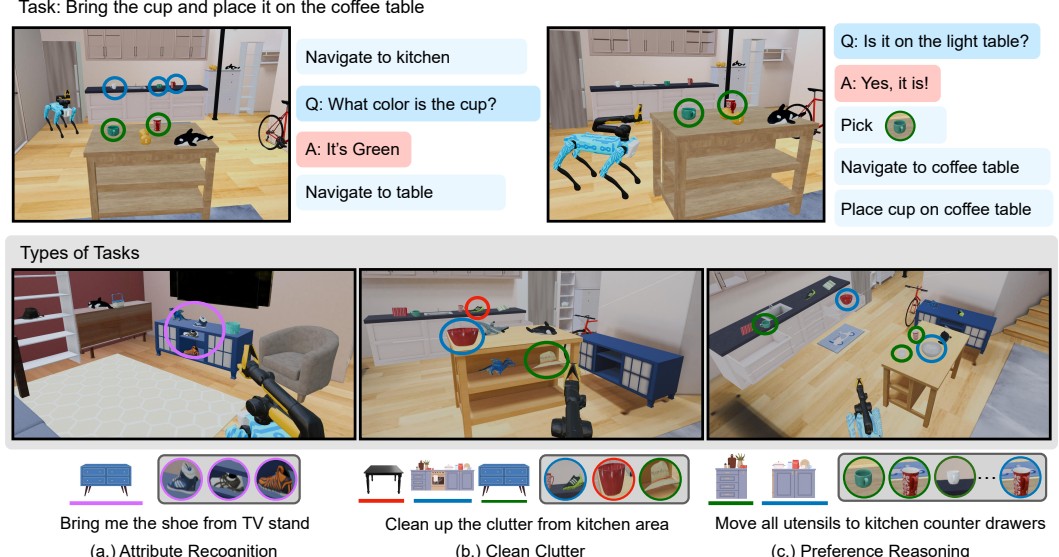

**Figure 1.** ASK-TO-ACT **task**. In this task, the user requests a specific green cup but, instead of describing it in detail, asks the agent, *"Bring the cup and place it on coffee table"*. Since the user's intent is unclear, an agent must ask a *minimum number* of clarification questions to disambiguate the requested object (*e.g. "Are you looking for a red cup?"* or *"Is it on the kitchen counter?"*). We consider under-specified single and multi-object rearrangement tasks that involve inquiring about user preferences and resolving different types of ambiguities, about object attributes, spatial relationships, object size, placement location, or combinations of the four.

demonstrations (Black et al., 2024; Ramrakhya et al., 2022; Kim et al., 2024; O'Neill et al., 2024; Ehsani et al., 2024) with low-level actions combined with imitation learning (IL) or use large-scale reinforcement learning (Wijmans et al., 2020; Andrychowicz et al., 2018; Zeng et al., 2024) (RL) with manually designed rewards. However, neither of these methods are trivial nor scalable, especially when training agents that require intermediate natural language communication: Gathering natural language interaction data interleaved with robot actions introduces additional overhead in already expensive and time-consuming robot teleoperation. Alternatively, training such agents with RL requires access to dense reward functions, which are impractical to manually design to encourage task-specific deductive or commonsense reasoning. Prior works (Ren et al., 2023; Huang et al., 2022; Jr. & Manocha, 2024; Min et al., 2024) focusing on similar settings take advantage of the prior knowledge and rich commonsense reasoning abilities of Large Language Models (LLMs) in a zero-shot manner for task planning and reasoning about ambiguity to ask clarification questions. However, these methods either require careful prompt engineering to prevent the robot from excessively relying on seeking assistance, or operate under strong assumptions, i.e. the environment is fully observable and can be represented in text without any errors. These assumptions are unrealistic – real-world environments impose partial observability, requiring agents to actively explore and gather information.

To address these limitations, our key idea is to bootstrap the learning signal required for training embodied agents capable of interacting and asking in an end-to-end manner by leveraging an LLM's contextual commonsense and deductive reasoning ability to resolve ambiguity. We accomplish this by adapting a multimodal large language model (MLLM) (Li et al., 2024; Liu et al., 2024) into a vision-language-action (VLA) model (Szot et al., 2024b; Black et al., 2024; Kim et al., 2024; Szot et al., 2024a) using large-scale RL with reward signal generated using an LLM. To distill reasoning ability required for resolving ambiguity into this VLA model, we propose to use per-step rewards generated using an LLM (Ma et al., 2024; Yu et al., 2023a) that evaluates both the actions taken and natural language question asked by the agent in context of the task. Through our experiments, we show that LLMs are highly effective at generating per-step rewards for such tasks that require interacting with environments and asking questions, when provided with the right representation of task and environment in text – information that can be easily curated at training time using privileged simulator state. Unlike prior works (Ma et al., 2024; Yu et al., 2023a) that use LLMs to generate code based rewards for low-level control policies, our work is one of the first ones to demonstrate LLMs can be used as reward model for tasks that require common sense and deductive reasoning in embodied settings.

To evaluate the effectiveness of our method we instantiate the ASK-TO-ACT task in a partial observable setup where agents process visual observations, and at each step output either actions or natural language questions to solve a rearrangement task (Weihs et al., 2021; Szot et al., 2021), in contrast to prior works (Ren et al., 2023; Jr. & Manocha, 2024). We set up ASK-TO-ACT in Habitat 3.0 (Puig et al., 2024) using 83 scenes from the ReplicaCAD Szot et al. (2021) dataset, with 42 object categories from Google Scanned Objects (Downs et al., 2022). We evaluate the generalization ability of these agents along two axes: 1) **Unseen Scenes**: where we evaluate on seen tasks in unseen scenes which include unseen configurations of furniture and objects, and 2) **Unseen Tasks**: where we evaluate the agent on novel task instances in unseen scenes *i.e.* novel compositions of the tasks with ambiguity described in Sec. 3 not seen by the agent during training.

We compare our proposed method against strong zero-shot baselines leveraging proprietary LLMs and MLLMs like GPT-4o (OpenAI, 2023), as well as methods for fine-tuning open-source MLLMs using synthetically generated supervised fine-tuning (SFT) data for the ASK-TO-ACT task. Our method, which employs large-scale RL with LLM-generated rewards to fine-tune an open-source MLLM, outperforms all baselines by a large margin, $10.4\%$ and $16.5\%$ for success rates on UNSEEN SCENES and UNSEEN TASKS evaluation splits. We also find that training VLA policies using only task subgoal rewards (*i.e.* no explicit rewards for asking the right questions), which can be programmatically written, achieve performance close to random on ASK-TO-ACT task. This indicates that training embodied agents to act and reason about ambiguity requires dense reward signals. Additionally, we analyze the behavior of our RL-trained policy and find that as the number of questions these agents can ask increases, the success rates of these policies improve and they generalize better to UNSEEN TASKS.

## 2 RELATED WORK

**Agents that ask for help.** Our work builds on prior works in embodied AI (Padmakumar et al., 2021; Thomason et al., 2019; Gao et al., 2022; Banerjee et al., 2020; Singh et al., 2022) which explore effective human-robot interaction through dialogue. In these settings, an embodied agent can ask questions in natural language to improve task performance. In ASK-TO-ACT, we study this in the context of multi-object rearrangement tasks with under-specified instructions where the agent needs to reason about ambiguity and identify user preferences by engaging in a multi-round dialogue (1-7 rounds). In recent works (Ren et al., 2023; Jr. & Manocha, 2024), the most common approach is to leverage an LLM's commonsense reasoning and language generation ability in a zero-shot manner to handle such human-robot interactions. However, these methods require extensive prompting and operate under assumption of full observability *i.e.* the complete error-free environment state is represented in text. We take a step towards relaxing these constraints and make the task setup more realistic by setting up the ASK-TO-ACT task in a partially observable setting.

**Reward generation using LLMs.** A promising method to train robot policies without expensive human demonstrations is to use RL with synthetically generated rewards (Ma et al., 2024; Yu et al., 2023b; Chen et al., 2025; Sarukkai et al., 2024; Ma et al., 2024). A common approach is to prompt the LLM to write code that takes symbolic features from the environment observations and produces a scalar output representing the reward. This approach has shown promising results on training policies for low-level robot control for complex dexterous manipulation, whole body control, locomotion, and object manipulation. A key difference of our work is that we are focused on teaching embodied agents ambiguity reasoning ability and language generation to ask for feedback using LLM generated rewards. Motivated by the effectiveness of using LLMs as reward models for complex math and code reasoning tasks (Zeng & Xu, 2023; Sarukkai et al., 2024; Xie et al., 2024; Chu et al., 2023), in our work we focus on evaluating the effectiveness of LLM reward models for embodied tasks that require reasoning about ambiguity. In our work, we show that this paradigm of training interactive policies using an LLM as a process reward model (PRM) (Setlur et al., 2025) is an effective approach for tasks that require reasoning about ambiguity, where writing rewards manually or collecting human demonstrations is not trivial or scalable.

## 3 ASK-TO-ACT TASK

Our goal is to build embodied agents capable of completing under-specified tasks by interacting with users to ask clarification questions to resolve ambiguity and identify preferences of a user in partially

observable settings. To study building such agents, we propose the ASK-TO-ACT task as shown in Fig. 1, where an agent is spawned randomly in an unseen indoor environment and tasked with fetching single or multiple instances of objects and placing them on a designated receptacle using an under-specified language instruction. Consider the example shown in Fig. 1: the user wants a specific green cup but, instead of describing it in detail, asks the agent, *"Bring the cup and place it on coffee table"*. While searching, the agent finds multiple cups on the kitchen counter and the table - two red cups, a white cup, and two green cups. Since the user's intent is unclear, an optimal agent must ask the fewest clarification questions to disambiguate the requested object. To study an agent's ability to reason about ambiguity and preferences of the user for such under-specified tasks, we synthetically create tasks of varying difficulty that require identifying user preferences or resolving different types of ambiguities, e.g. about variations in object appearance, location, placement location, size or composition of these axes. Fig. 1 and Fig. 5a shows examples of tasks in our dataset, and below we describe each task and evaluation criteria in detail:

1.) **No Ambiguity**: These include single or multi-object rearrangement instructions that clearly specify the target objects and do not require reasoning about ambiguity or user preferences.

2.) **Attribute Recognition**: Agent must reason about appearance attributes like color or category to disambiguate target objects (e.g., blue shoe, white shoe *vs*. orange shoe as shown in Fig. 1 (a)).

3.) **Spatial Reasoning**: Agent must differentiate between objects with similar appearance based on spatial location to identify target object (e.g., red cup on the light table, or kitchen counter, or dark table as shown in Fig. 5a (b) in Appendix A).

4.) **Object Size**: The agent must distinguish between objects of different sizes but similar appearance and location to identify target object (e.g., a large red bowl vs. a small red bowl, shown in Fig. 5a (e) in Appendix A).

5.) **Compositional Ambiguity Reasoning**: A combination of attribute recognition, object size, and spatial reasoning, where the agent must consider appearance, geometry and location to ask minimum number of clarification questions to identify the target object (as shown by example in Fig. 5a (c, f) in Appendix A).

6.) **Clean Clutter**: Multi-object rearrangement tasks with under-specified instructions, where the agent must determine which objects on a source receptacle are "clutter" and place them in appropriate target locations. In each instance, the user does not know the full set of objects on the source receptacle, and only a subset of the objects present are considered clutter. As a result, the agent must communicate the current state of the receptacle (see example in Fig. 1 (b)). Additionally, the user has hidden placement preferences for each object category (e.g., cups should go in the left drawer of the kitchen counter, while plates belong in the top cabinet).

7.) **Preference based**: Multi-object rearrangement tasks that require identifying user preferences of placement location for multiple objects of a semantic category. For example, "Move all utensils to kitchen" requires asking user for preference of target receptacle for each object categorized as utensils (*e.g.* cups go on top drawer and bowls go on right drawer of kitchen counter, as shown in Fig. 1 (c)).

**Simulation Answers.** Evaluating questions asked by an agent in context of a task in ASK-TO-ACT in a programmatic manner is important for simulating diverse users preferences and language responses. To do so, we use an LLM to build an answering module for ASK-TO-ACT task which takes as input the task instruction, privileged information of environment state represented as text, metadata of the target objects a user is requesting and expected target location, and is tasked with answering questions generated by the embodied agent. Details about the prompts to answering LLM are in Appendix C

**Evaluation Splits.** We assess the generalization ability of these agents along two axes: (1.) **Unseen Scenes**: Evaluates how well the agent can perform tasks under new object layouts in the scene, or new instances of previously seen object classes. (2.) **Unseen Tasks**: Tests the agent's ability to handle novel instances of ambiguity in unseen scenes, requiring the agent to ask more questions than training tasks. For example, for attribute recognition tasks, in training the agent only needs to distinguish between 2 objects of different attributes but in UNSEEN TASKS split it needs to distinguish between 3 or 4 instances. See Appendix A for examples.

**Task Setup and Agent Action Space.** We instantiate the task using Habitat 3.0 (Puig et al., 2024) in 63 training scenes and 20 evaluation scenes from the ReplicaCAD (Szot et al., 2021) dataset

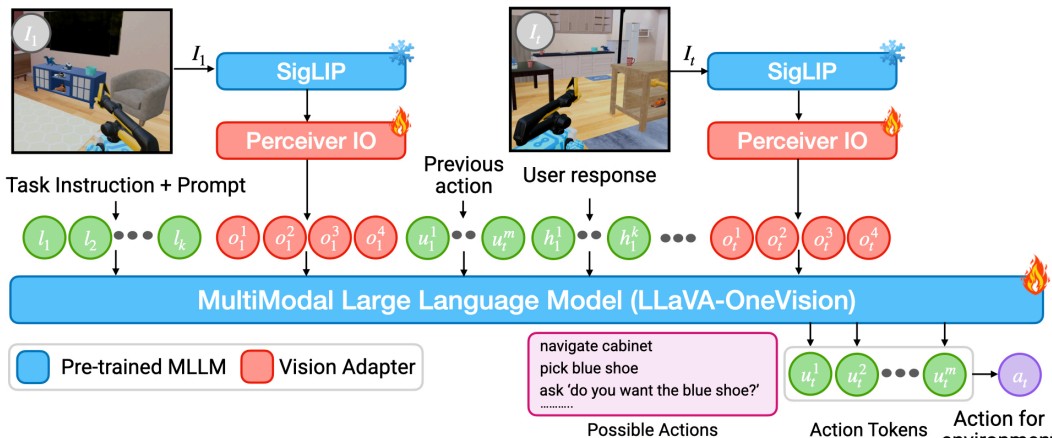

**Figure 2. MLLM Policy Architecture**. The policy takes as input a task instruction, past observations, actions, user response to questions asked and outputs a high-level action or a question in natural language.

with 42 object categories from Google Scanned Objects (Downs et al., 2022). We use the Spot robot embodiment for the ASK-TO-ACT task with a height of 1.41m and radius of 0.25cm. At each timestep the agent has access to a $480 \times 640$ resolution RGB image, robot joint positions, an indicator of whether the robot is holding an object or not, relative base egomotion, and task instruction. The agent's action space is implemented in Habitat 3.0 (Puig et al., 2024) with a set of oracle low-level skills, and focus on high-level decision-making. Low-level skills include picking objects by name, placing on receptacles, navigating to specific locations, and opening articulated receptacles. Example action outputs include pick(apple), place(sink), navigate(cabinet), open(left drawer), and ask_question("Do you want the red cup?"). At each timestep, the policy selects from 125 predefined skills or an open-ended "ask question" action to generate a natural language query when clarification is needed.

## 4 APPROACH

Our method, called AUTOASK, adapts a pre-trained multi-modal large language model (MLLM) to a vision-language-action (VLA) model capable of simultaneously taking actions, and generating natural language questions to resolve ambiguity in multi-modal embodied decision-making settings – without relying on human annotations or hand-engineered rewards. We use reinforcement learning (RL) with LLM-generated rewards to train a VLA for ASK-TO-ACT.

### 4.1 POLICY ARCHITECTURE

We adapt the LLaVA-OneVision (Li et al., 2024) MLLM architecture to a VLA model for task-planning and asking questions as illustrated in Fig. 2. The MLLM policy takes as input a task instruction along with a sequence of past observations, actions, user response to questions asked and outputs a high-level action (*i.e.* skill) or a question in natural language. As shown in Fig. 2, the task instruction is first encoded into language tokens. Traditional MLLMs embed a single visual input into a large number of visual tokens *e.g.* LLaVA-OneVision (Li et al., 2024) uses 729 tokens. However, in partially observable embodied settings these agents need to reason over a long sequence of past observations to complete a task. Using a large number of visual tokens for each observation would quickly increase the context lengths (*e.g.* 32 visual observations $\times 729$ tokens $= \sim 23k$ tokens). To make it feasible to train MLLMs with a long observation history, we employ a Perceiver model (Jaegle et al., 2022) to downsample the visual tokens per observation (to $k = 4$ in our case), similar to Szot et al. (2024a). Next, these downsampled visual tokens from the current observation $o_t$ are interleaved with task instruction tokens $(l_1, ..., l_k)$, text representation of past actions $(u_{t-1}^1, .., u_{t-1}^m)$, and language response given by user for questions asked at each step tokenized $(h_{t-1}^1, .., h_{t-1}^n)$. These interleaved language and visual tokens are processed by the MLLM, which autoregressively predicts a sequence of language tokens corresponding to a discrete action or a natural language question. Each agent action $a \in A$ (a skill or possible questions $q \in Q$)

is represented as free-form text sequence, tokenized as $(u_t^1, ..., u_t^m)$. Sequences corresponding to invalid actions are treated as a no-op to prevent execution errors.

## 4.2 TRAINING WITH REINFORCEMENT LEARNING USING LLM-GENERATED REWARDS

To finetune LLaVA-OneVision (Li et al., 2024) base MLLM with embodied data we use online RL with LLM-generated rewards for the ASK-TO-ACT task. We show that current state-of-the-art LLMs can be used to generate reward signal for RL training, when provided with task and environment information in the right representation. Specifically, when the agent asks a question we provide an LLM with privileged environment state, task instruction, metadata of all target objects, expected target location, questions asked so far and action at current step in text representation. Next, we ask the LLM to generate a binary reward that evaluates whether the question asked helps resolve ambiguity or identify user preference compared to information from the previous step. See Appendix C for more details and the prompt. Eq. (1) describes the reward function we use, which consists of a sparse reward $\mathbb{1}_{\text{success}}$ for completing the task, subgoal rewards $\mathbb{1}_{\text{subgoal}}$ generated by the LLM for achieving intermediate objectives (*e.g.* picking target object, navigating to target location, etc), a ckck penalty $r_5$ at every timestep to encourage task completion in fewer steps, and a reward for asking clarification questions $\mathbb{1}_{\text{useful\_question}}$ that aids in disambiguating the target object. The reward at step $t$ is computed as follows:

$$
\begin{aligned}
r_t =& r_1 \cdot \mathbb{1}_{\text{success}} + r_2 \cdot \mathbb{1}_{\text{subgoal}} + r_3 \cdot \mathbb{1}_{\text{useful\_question}} \\
& - r_4 \cdot \mathbb{1}_{\text{exceed\_budget}} - r_5,
\end{aligned}
\tag{1}
$$

where $\mathbb{1}_{\text{useful\_question}}$ (computed using an LLM as a reward model) indicates if a question asked by the agent is valid and helps make progress towards disambiguating the target object or not. Consider the example of placing utensils in kitchen, if the environment has 2 cups and 1 plates on a table and the agent asks "Where should I place cups?" and if the user responds "left drawer of kitchen counter", then the agent should ask "Where should I move the plates?" to make progress towards the task instead of asking "Is the plate on the table?". When training policies under a budget of questions we penalize the agent for every question that exceeds a pre-specified budget given by $\mathbb{1}_{\text{exceed\_budget}}$. By default the budget is set to the minimum number of required questions $K$ to solve the task.

**Implementation details**. We train the MLLM policy using DD-PPO (Wijmans et al., 2020), an adaptation of PPO (Schulman et al., 2017) for distributed training, for 50 million steps on 8 A40 GPUs. To generate rewards for our task we use the Llama-3 (Grattafiori et al., 2024) LLM. The LLM reward model runs as a separate process using vLLM (Kwon et al., 2023) and processes reward labeling requests asynchronously. We provide training details and prompts in Appendix B and Appendix F.

## 5 EXPERIMENTS

In this section, we compare our method (AUTOASK) of adapting MLLMs as VLA using RL with LLM-generated rewards to proprietary MLLMs in a zero-shot setting and MLLMs trained using synthetically generated SFT data for our task.

### 5.1 BASELINES

For methods involving MLLM fine-tuning, we use LLaVA-OneVision 0.5B (Li et al., 2024) as our base MLLM with modifications described in Sec. 4.1. We use the 0.5B model due to compute constraints for running RL training on A40 GPUs. Below, we outline the details of each baseline:

**(a.) Fully Observable Text WorldGraph + ReAct (Zero-shot)**. We provide GPT4o with a fully observable text-based world graph describing the environment, including receptacles, objects, and their locations (*e.g.* "The apple is on the coffee table."). Additionally, the LLM is equipped with a skill library for executing actions, a history of previous actions, and the task instruction. At each timestep, the LLM generates a reasoning chain via ReAct (Yao et al., 2023) prompting followed by an action to complete the task. By eliminating perception limitations, this baseline evaluates the zero-shot planning and reasoning capabilities of LLMs for solving under-specified embodied tasks.

| Method | Full Obs. | UNSEEN SCENES | | | UNSEEN TASKS | | |
|---|---|---|---|---|---|---|---|
| | | SR ($\uparrow$) | ARS ($\uparrow$) | QR ($\downarrow$) | SR ($\uparrow$) | ARS ($\uparrow$) | QR ($\downarrow$) |
| 1) WG + ReAct* | ✓ | 41.7 | 38.9 | 3.3 | 39.9 | 37.1 | 3.2 |
| 2) WG + ReAct* (FS) | ✓ | 46.7 | 44.7 | 0.7 | 44.4 | 42.6 | 1.2 |
| 3) WG + ReAct* (FS) | ✗ | 35.5 | 26.6 | 3.9 | 32.2 | 26.4 | 3.8 |
| 4) GPT4o + SoM + ReAct | ✗ | 15.7 | 13.5 | 1.9 | 14.3 | 12.7 | 2.1 |
| 5) LLaVA-OV SFT | ✗ | 33.6 | 26.6 | **0.8** | 29.5 | 25.1 | **1.1** |
| 6) AUTOASK (Ours) | ✗ | **45.9** | **40.6** | 0.9 | **38.7** | **35.2** | 1.5 |

**Table 1. Results**. Evaluation of all methods on UNSEEN SCENES and UNSEEN TASKS evaluation splits. FS denotes few-shot examples, * denotes access to privileged information, Full Obs. stands for full observability.

**(b.) Fully Observable Text WorldGraph + ReAct (Few-shot)**. This baseline extends (a) by providing the LLM with a few in-context examples that demonstrate task planning and ambiguity resolution strategies in ASK-TO-ACT to assesses whether in-context learning improves task performance.

**(c.) Partially Observable Text WorldGraph + ReAct (Few-shot)**. While baselines (a) and (b) assume privileged access to a fully observable world graph, constructing such representations in real-world settings is often infeasible. This baseline relaxes that assumption by providing the LLM with a partially observable text-based world graph requiring LLM agent to actively explore to solve the task.

**(d.) Vision GPT4o + SoM + ReAct**. Building an error-free text representation of real-world environments is challenging. This baseline evaluates whether existing MLLMs can solve the ASK-TO-ACT task using egocentric visual observations. At each timestep, MLLM receives the robot's visual input annotated with Set-of-Marks (SoM) (Yang et al., 2023) along with a skill library and a memory module that maintains textual history of past observations and actions.

**(e.) LLaVA-OneVision SFT**. Generating training data for SFT is non-trivial, as the agent must first explore the environment to locate relevant objects, ask contextually grounded questions at the right time to resolve ambiguity and identify user preferences, and finally execute the task. To achieve this, we design an expert using heuristic exploration and an LLM. The agent first explores the scene using frontier exploration (Yamauchi, 1997) until all receptacles are seen, then converts its observations into a text-based world graph. This graph, along with the task instruction and privileged information about the target objects, is passed to an LLM, which generates a sequence of clarifying questions to identify target objects. Next, a heuristic planner executes the rearrangement tasks to generate the final trajectory which consists of exploration actions, LLM-generated questions as actions, and actions taken to rearrange objects. Only successful trajectories are included in training. Our SFT dataset contains ∼40k trajectories in 63 training scenes in the ReplicaCAD dataset.

## 5.2 METRICS

We report three metrics: 1.) **Success Rate (SR)**: a measure of successful task completion, 2.) **Ambiguity-Resolution Efficiency Score (ARS)**: a measure for the agent's ability to successfully complete task while asking *minimum* number of required clarification questions and penalizing irrelevant ones. ARS is computed using the following function:

$$ARS = \frac{\mathbb{1}_{\text{success}}}{1 + \text{abs}(q_{\text{relevant}} - K) + q_{\text{irrelevant}}}, \tag{2}$$

where $\mathbb{1}_{\text{success}}$ denotes whether an episode was successful, $q_{\text{relevant}}$ is the number of relevant questions asked per episode, $q_{\text{irrelevant}}$ is the number of irrelevant/redundant questions asked per episode, and $K$ denotes the minimum number of questions required to resolve ambiguity for each episode. ARS penalizes an agent equally for asking fewer or more questions than minimum required questions to resolve ambiguity for each episode equally. 3.) **Question Ratio (QR)**: ratio of total questions agent asked *vs.* the minimum required questions to solve the task *i.e.* $(q_{\text{relevant}} + q_{\text{irrelevant}})/K$.

## 5.3 RESULTS

Tab. 1 presents the results of evaluating all methods on UNSEEN SCENES and UNSEEN TASKS splits. To establish an upper bound on performance, we first evaluate zero-shot LLMs with privileged information (rows 1 and 2 of Tab. 1). When an LLM is provided with a fully observable text-based

|  | UNSEEN SCENES | | | UNSEEN TASKS | | |
|---|---|---|---|---|---|---|
| Method | SR (↑) | ARS (↑) | QR (↓) | SR (↑) | ARS (↑) | QR (↓) |
| 1) Success Reward | 0.0 | 0.0 | 0.0 | 0.0 | 0.0 | 0.0 |
| 2) Subgoal Reward | 22.4 | 21.6 | **0.4** | 16.5 | 6.9 | **0.5** |
| 3) AUTOASK | **45.9** | **40.6** | 0.9 | **38.7** | **35.2** | 1.5 |

**Table 2. Reward Choice**. Evaluation results of using different rewards for training our method on UNSEEN SCENES and UNSEEN TASKS evaluation splits.

representation of the environment and uses ReAct (Yao et al., 2023), it achieves a success rate of 41.7% and 39.9% and ARS of 38.9% and 37.1% on UNSEEN SCENES and UNSEEN TASKS splits. This highlights the inherent difficulty of task planning and ambiguity resolution for LLMs in a zero-shot setting. Next, we augment LLMs with privileged in-context examples demonstrating how to perform the ASK-TO-ACT tasks. This significantly improves performance, increasing success rate to 46.7% and 44.4% on unseen scenes and tasks. With in-context examples we find LLMs perform quite well on single-object rearrangement tasks but struggle with multi-object rearrangement tasks where interaction with articulated objects is required. We then evaluate the same baseline under partial observability (row 3), where the agent does not have access to a fully observable world graph but still has in-context examples. This leads to an absolute drop in success rate by 11.2% and 12.2% on both splits (row 3 *vs.* row 2), indicating that task planning and ambiguity resolution become significantly more challenging when the agent must actively explore to gather information. As shown in Tab. 1, our approach AUTOASK (row 6), which fine-tunes a MLLM (Li et al., 2024) using RL with LLM-generated rewards, outperforms all methods operating under partial observability and achieves 45.9% and 38.7% on success rate and 40.6% and 35.2% ARS on UNSEEN SCENES and UNSEEN TASKS splits. AUTOASK outperforms the policy trained with synthetic SFT data (row 5 *vs.* row 6) by absolute 6.5 − 12.3% on success rate and 8.8 − 20.0% on ARS. We find agents trained with SFT tend to be more conservative (lower than 1 QR as shown in row 5), asking fewer questions compared to RL-trained counterparts which could be the reason for difference in performance, additional analysis shown in Sec. 5.4. Additionally, AUTOASK also surpasses a strong zero-shot baseline (row 4) that uses vision GPT4o (OpenAI, 2023) with SoM (Yang et al., 2023) prompting and a text-based history of past observations by an absolute margin of 24.4 − 30.2% on success rate. These results demonstrate online RL combined with LLM-generated rewards is an effective method for training agents that can interleave acting with asking relevant clarification questions to resolve ambiguity.

**Choice of reward for RL training**. In Tab. 2, we ablate the choice of reward function for RL training to demonstrate effectiveness of using an LLM to generate rewards for ASK-TO-ACT. We consider two simple rewards that can be programmatically defined for our task: (1.) Success reward: a sparse success reward of value 10 only given a at the end of the episode if the agent succeeds and 0 otherwise. (2.) Subgoal reward: the agent is rewarded for accomplishing any subgoal $\mathbb{1}_{subgoal}$ that are required to completing the overall task as defined in Eq. (1) in addition to a success reward. This reward can be programmatically generated for training episodes with access to privileged information from simulator for each episode. Note, this reward does not provide dense step-by-step incentive for exploration or to ask questions to resolve ambiguity; instead, it implicitly rewards the agent to ask questions that lead to subgoal success which makes RL training difficult. As shown in Tab. 2, neither of these rewards are sufficient to learn an effective MLLM policy for our task, demonstrating the need for precise per-step rewards that incentivize agents to not only act but also ask meaningful questions.

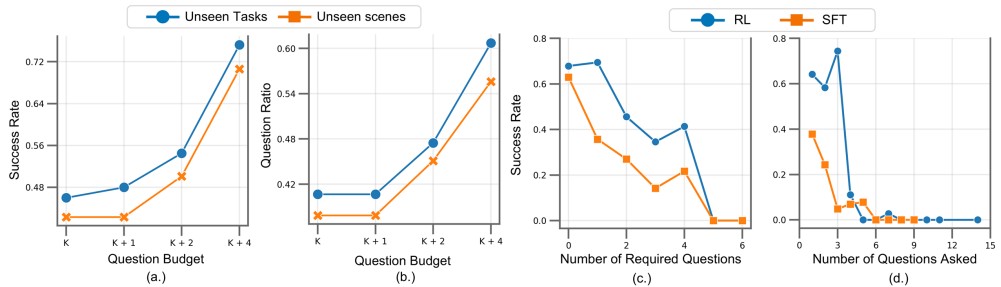

**Figure 3. Analysis**. (a.) Task Performance vs. Budget of Questions. Evaluation performance of policies trained under different budget of questions *vs.* task Success Rate and Ambiguity Resolution Efficiency score. (c.) Success Rate vs. Number of Required and Asked Questions. Evaluation performance of SFT and RL trained policies *vs.* number of required and total questions asked by the agent.

**Training policies under variable budget**. A desired skill for an embodied agent that can ask questions is to adhere to user's preferences about how often they would like a robot to ask clarification questions. Some would prefer an agent ask as few questions as possible for better user experience by trading off task success rates. In contrast, some users would be fine with an agent asking as many questions as it would like to ensure task success rates are higher. Motivated by this, we train multiple MLLM policies using RL with LLM generated rewards with a variable upper-bound on maximum number of questions an agent can ask. Specifically, we train policies with a budget of $B \in \{K, K+1, K+2, K+4\}$ questions, where $K$ is minimum required question for a task in ASK-TO-ACT dataset. The agent can ask at most $B$ questions in a single episode (either relevant or irrelevant) without incurring any penalties. Note, for this experiment $B$ is either equal to $K$ *i.e.* ask as close to minimum required questions as possible or can be quite high $K+4$ where an agent can ask as many as 4 extra questions than minimum required in each episode without incurring any penalties. Additionally, the agent will only be rewarded for relevant questions from all questions it asked. We only penalize the agent for each question asked after exceeding the question budget $B$. Fig. 3 (a.) and (b.) shows success rates and question ratio of policies trained with different budgets under the reward setting described in Eq. (1). As shown in Fig. 3 (a.), as we increase the number of questions the agent can ask, the success rates increase; however, there is a clear trade-off between increase in success rates and question ratio (*i.e.* asked questions to minimum required), see Fig. 3 (b.).

## 5.4 ANALYSIS

**Quantitative analysis of agent behavior**. In Fig. 3 (c.), we show the performance of MLLM policies trained using RL and SFT *vs.* minimum required questions to solve tasks on UNSEEN TASKS split. These results show, as the number of required questions for a novel task increases the performance of both methods drops. In Fig. 3 (d.), we show the performance of these policies *vs.* number of questions asked by the agents. This shows, as the number of questions asked by RL agents increase (from $1-3$ questions) the task success rate starts increasing. We also observe, agents trained with SFT tend to be more conservative asking fewer questions compared to RL-trained counterparts that could explain the difference in performance as shown in Tab. 1 (row 5 *vs.* 6).

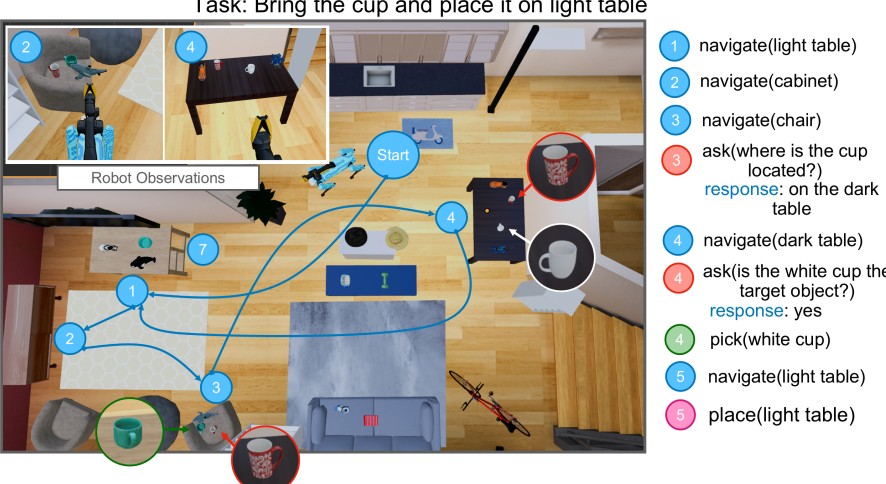

**Figure 4. Qualitative Example.** Successful trajectory on an evaluation episode from UNSEEN TASKS split.

**Qualitative examples**. Fig. 4 shows a successful example of our method on UNSEEN TASKS split. The agent first explores until it finds all 4 instances of the cup then it asks a sequence of questions. Finally, based on responses from the user it successfully brings the green cup user wants to the light table. For more qualitative examples, see Appendix D.

**Answerer and AUTOASK Reward Analysis**. We present analysis of the accuracy and robustness of using an LLM as an answerer and the reward model for the ASK-TO-ACT task in Tab. 3. Specifically, we conduct human verification using 3 LLMs as an answering module and a reward model for 100 unique task instances covering 7 task categories in Ask-to-Act. For each task, we first generate 1-3 different solutions that disambiguate target objects by asking a varying number of questions

| | ANSWERER | REWARD MODEL | | |
|---|---|---|---|---|
| LLM | Accuracy (↑) | Accuracy (↑) | Precision (↑) | Recall (↓) |
| 1) GPT-4o Reward | 94.5 | 77.9 | 68.3 | 85.4 |
| 2) Llama 3.1 8B | 85.5 | 66.4 | 54.9 | 70.0 |
| 3) Llama 3.1 70B | 90.2 | 71.4 | 63.4 | 77.7 |

**Table 3. Answerer and Reward Model Analysis**. Human verification analysis for using LLMs as answering module and reward model for ASK-TO-ACT on question solutions for 100 task instances from evaluation splits.

| Failure Mode | % Episodes |
|---|---|
| 1) Over Questioning | 65.4 |
| 2) Under Questioning | 34.8 |
| 3) Irrelevant Questions | 10.2 |
| 4) Task Planning | 7.7 |

**Table 4. Failure Mode Analysis**. Categorization of failure modes of AUTOASK policy evaluated on ASK-TO-ACT evaluation splits.

from 1-5 through heuristics. Through this approach we end up with 200 unique question solutions. Out of the 200 generated solutions we ensure ∼40% are failures *i.e.* include questions that do not disambiguate target objects and receptacles successfully to ensure we are evaluating both positive and negative scenarios that occur during training. Using these solutions we first evaluate answering module accuracy by asking a human verifier to annotate whether the language responses generated by the LLM answering module are correct or not for every individual question. Similarly, we also generate responses from the LLM reward models for all the question solutions and ask the human verifier to annotate whether the reward generated by the LLM reward model for each question in the solution is correct or not. A question solution (*i.e.* sequence of questions for a task) is deemed successful only if the reward model labels reward for each question successfully.

As shown in Tab. 3, LLM as answering module are not perfect answerer which naturally leads to instances where the LLM answer has noise or miscommunication. For example, consider the task shown in Fig. 1, if the agent asks a question '*Is it on the light table?*' as the first question the LLM answering module is unable to identify what does the agent refer to by asking '*Is it*'. Due to this the answerer replies '*No, it is not*' which leads to agent failing to complete the task. In contrast, if the agent first asks '*What color is the target cup?*' followed by '*Is it on the light table?*', the LLM answering module is able to associate '*Is it*' to the '*Green cup*' and correctly answer whether it is on the light table or not. We find this is a common failure mode of our LLM answerer.

**Failure Mode Analysis**. We present an analysis of categorizing different failure modes of policy trained using AUTOASK in Tab. 4. We find the policy exhibits 4 failure modes: (1.) Over Questioning: Asking more than $K$ questions which include redundant follow-ups for same information and failing to complete task, (2.) Under Questioning: Asking less than required $K$ questions and rearranging incorrect target object, (3.) Irrelevant Questions: Asking minimum required questions $K$ but a subset of these questions are not relevant to disambiguating the target object *i.e.* not useful, (4.) Task Planning: Asking right questions but failing to complete the task after disambiguating the target object details.

## 6 CONCLUSION

In this work we introduce ASK-TO-ACT, a novel task where an embodied agent is tasked with fetching a specific instance of an object through an underspecified language instruction. To solve this task, the agent needs to be capable of reasoning about ambiguity based on the context of the environment and ask clarification questions in natural language to resolve the ambiguity. To train such agents, we propose an approach that adapts multimodal large language models (MLLMs) using reinforcement learning (RL) with LLM-generated rewards. Our results demonstrate that this approach significantly improves task success rates and ambiguity resolution efficiency score, outperforming strong zero-shot baselines using GPT-4o and open-source MLLMs fine-tuned with SFT on LLM-generated data. We evaluated our method on both unseen scenes and unseen tasks, showing that an RL-trained VLA model can generalize effectively to novel object arrangements and ambiguous task instructions. Our findings highlight effectiveness of dense, context-aware LLM-generated rewards for training embodied agents capable of resolving ambiguity by interacting in natural language.

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

*white basket, green basket, brown basket, white bowl, red bowl, green bowl, red cup, green cup, white cup, white casserole, green casserole, black casserole, green cushion, yellow cushion, blue cushion, yellow dumbbell, green dumbbell, blue dumbbell, blue jug, orange jug, white jug, blue towel, red towel, green towel, grey hat, yellow hat, white plate, green plate, pink plate, black shoe, blue shoe, orange shoe, grey toy, blue toy, black toy, red vase, blue vase, pink vase, black teapot, white teapot, yellow teapot*

**Table 5.** Full list of object categories used for ASK-TO-ACT task.

## A DATASET

In Fig. 5a, we show additional examples of different types of ambiguities in our ASK-TO-ACT dataset. Specifically we show 3 additional examples for:

- **Attribute Recognition**: The agent must reason about appearance-based attributes such as color and object category to disambiguate the object user is looking for (e.g., a red bowl and white bowl as shown in Fig. 5a (a)).

- **Object Size**: The agent must distinguish between objects of different sizes but similar appearance and location to disambiguate the object user is looking for (e.g., a large red bowl vs. a small red bowl as shown in Fig. 5a (b)) .

- **Attribute/Spatial Reasoning and Object Size**: A combination of attribute and spatial reasoning with object size, where the agent must account for both appearance or location and size to effectively disambiguate the target object the user is looking for. For the example shown in Fig. 5a (c), the agent needs to reason about different colored shoes (blue, white, and orange shoe) which are located on the blue cabinet and the sofa and some of those have different size *e.g.* large blue shoe on blue cabinet *vs.* small blue shoe on sofa and large white shoe on sofa *vs.* small white shoe on blue cabinet.

**Object Categories**. We present the full list of object categories used for ASK-TO-ACT dataset in Tab. 5.

## B TRAINING DETAILS

**Problem Setup.** Our problem setup can be formulated as a Partially-Observable Markov Decision Process (POMDP), defined by a tuple $(S, O, A, P, R, p_0, \gamma)$ where $S$ is state space, $O$ is observation space, $A$ is action space, $P$ represents transition dynamics, $R$ is reward function, $p_0$ is initial state distribution and $\gamma$ is discount factor. In our setting, $O$ is a combination of responses from the user (for any questions asked) and visual observations, which come from the robots egocentric RGB camera, and provide only partial views of the environment. We consider the extension of including a goal distribution $G$ and the case where the reward is formulated as $R(s, g)$ for $s \in S$ and $g \in G$. We aim to learn a language-conditioned policy $\pi(a|o, g)$ mapping from observation $o$ and task instruction $g$ to an action $a$ that maximizes the sum of discounted rewards $\mathbb{E}_{s_0 \sim p_0, g \sim G} \sum_t \gamma^t R(s_t, g)$.

**Training Details**. To train our policy using reinforcement learning (RL) we use PPO with Generalized Advantage Estimation (GAE) (Schulman et al., 2016). We use a discount factor $\gamma$ of 0.99 and set GAE parameter $\tau$ to 0.95. We do not use normalized advantages. To parallelize training, we use DD-PPO (Wijmans et al., 2020), an adaptation of PPO (Schulman et al., 2017) for distributed training, with 24 environments each on 8 A40 GPUs for 50 million steps . Each worker collects 24 frames of experience from 24 environments parallely and then performs 2 epochs of PPO update with 2 mini batches in each epoch. Tab. 6 details the default hyperparameters used in all of our training runs.

**Per-Step Reward Inference**. For per-step reward inference for evaluating questions asked by an agent, we use a vLLM (Kwon et al., 2023) server to host our LLM answer evaluation model using Llama-3.1 8B Instruct (Grattafiori et al., 2024) model. At each rollout step during RL training and policy evaluation, if the agent outputs a 'ask_question' action we send a request to the vLLM server and feed the response back to the policy at the next step. In order to achieve a high throughput we run 8 parallel vLLM servers and randomly send requests to different workers from training and evaluation workers.

**Token Normalization for Action Probability and Entropy**. When using MLLMs for RL training, each action $a \in A$ at every timestep is represented using a set of tokens $(u_t^1, ..., u_t^m)$. However, the

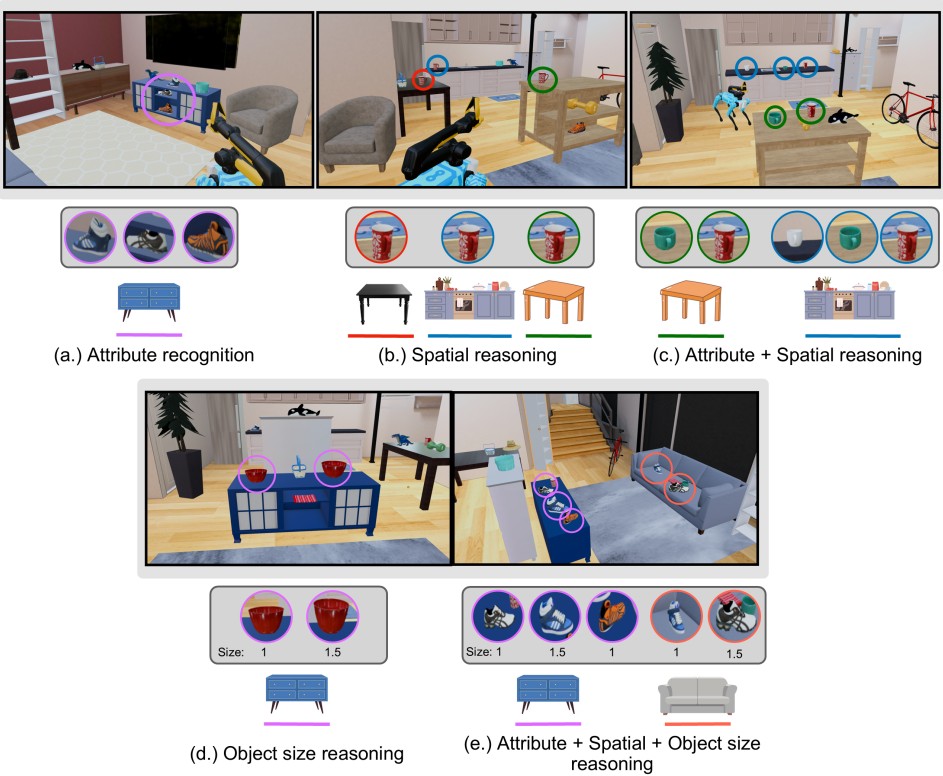

(a.) Attribute recognition
(b.) Spatial reasoning
(c.) Attribute + Spatial reasoning

(d.) Object size reasoning
(e.) Attribute + Spatial + Object size reasoning

**(a) Dataset**. Examples of different types of synthetically created ambiguities in ASK-TO-ACT tasks.

| Parameter | Value |
|---|---|
| Number of GPUs | 8 |
| Number of environments per GPU | 24 |
| Rollout length | 24 |
| PPO epochs | 2 |
| Number of mini-batches per epoch | 2 |
| LR | $2.5e^{-4}$ |
| Optimizer | Adam |
| Weight decay | 0.0 |
| Epsilon | $1.0 \times 10^{-5}$ |
| PPO clip | 0.2 |
| Generalized advantage estimation | True |
| $\gamma$ | 0.99 |
| $\tau$ | 0.95 |
| Value loss coefficient | 0.5 |
| Max gradient norm | 0.2 |
| DDPPO sync fraction | 0.6 |

**Table 6.** Hyperparameters used for RL finetuning.

MLLM only outputs token level probabilities which is different from probability of executing an action in the environment as used in RL traditionally. The token-level probability for each action $a_t$ can be represented by:

$$P_{\text{token}}(a_t|s) = \prod_{i=1}^{m} P(u_t^i|s, u_t^1, \dots, u_k^{i-1}) \tag{3}$$

To compute action probabilities, one naive approach is to take a sum over all token-level probabilities. When training MLLMs with action-space with variable number of tokens across actions one issue in Eq. (3), is that actions with larger number of tokens tend to have lower token-level probabilities, even though they might be more reasonable to take at some point in the environment. This issue happens because the probability of each token $P(u_t^i|\cdot)$ is always less than $1$. This could be problematic in RL optimization especially for the case when the agent needs to output a natural language question for ASK-TO-ACT task, simply because the questions tend to have more tokens than pre-defined skills represented in language. To remedy this issue, we use action length normalization technique normalize the token-level probabilities of the actions with the number of action tokens, which can be defined as:

$$logP(a_t|s) = \sum_{i=1}^{m} logP_{\text{token}}(u_t^i|s, u_t^1, \dots, u_k^{i-1})/m \tag{4}$$

We find action-length normalization helps stabilize training when working with our variable length action-space when using constrained grammar decoding (Park et al., 2025) for RL training.

## C   LLM REWARD GENERATION

Tab. 8 shows the prompt used by the question evaluation LLM used for ASK-TO-ACT task evaluation and RL training. In addition, we use the prompt 7 for generating optimal answer sequences and subgoals required to generate the SFT data and task subgoals described in Eq. (1) in Sec. 4.2.

$$
\begin{aligned}
r_t = & 10 \cdot \mathbb{1}_{\text{success}} + 2.5 \cdot \mathbb{1}_{\text{subgoal}} + r_3 \cdot \mathbb{1}_{\text{useful\_question}} \\
& - 0.05 \cdot \mathbb{1}_{\text{exceed\_budget}} - 0.01,
\end{aligned}
\tag{5}
$$

**Reward Details**. Eq. (5) shows the reward function we use with coefficients for each term. In this reward, $\mathbb{1}_{\text{success}}$ indicates whether the task was successfully completed and $\mathbb{1}_{\text{subgoal}}$ (generated by the LLM) indicates if the agent completed any subgoal required to complete the overall task. For example, for a task "Bring me the cup and place it on the coffee table", the agent needs to first search for all cups, pick the correct cup, then navigate to the coffee table, and finally place it. Similarly, $\mathbb{1}_{\text{useful\_question}}$ (generated by the LLM) indicates if a question asked by the agent is valid and helps make progress towards disambiguating the target object or not. Consider the example of fetching the cup, if the environment has $4$ cups on a table and the agent asks "where is it the cup?" and if user responds 'on the coffee table', then the agent should ask 'Is it the blue cup?' or 'Is it the yellow cup?' to find the target object instead of asking 'Is it blue cup in the sink?' to make progress towards solving the task. For each useful question $\mathbb{1}_{\text{useful}}$ the agent asks in an episode it is given a reward $r_3$ until total number of questions asked are less than the question budget. For example, if the task requires 3 questions then the agent will get $r_3 = 0.5$ for each relevant question it asks as evaluated by the LLM and $0$ otherwise or if the question budget is exceeded. When training policies under a budget of questions we penalize the agent for every question that exceeds pre-specified budget given by $\mathbb{1}_{\text{exceed\_budget}}$. By default the budget is set to minimum number of required questions $K$ to solve the task.

## D   QUALITATIVE EXAMPLES

We present additional qualitative examples of evaluating our method on UNSEEN TASKS evaluation split of ASK-TO-ACT dataset in Fig. 6.

Imagine you are a household robot. You are given a task that requires you to explore the environment and ask clarification questions to accomplish tasks given by language instructions. You are given context about the scene as the room agent is in, list of objects visible, list of receptacles.

For each task given as a language instruction you have to output a sequence of actions that the agent should take. These actions can include a clarification question as well. Ask a question only when required. For each action also output the reason to take the action with it. Actions can be one of the following:

1. nav(receptacle)
2. pick(object)
3. place(receptacle)
4. open(receptacle) // Strictly use when the object needs be to picked from or placed inside an articulated receptacle after navigating to it
5. close(receptacle) // Strictly use to close articulated receptacle after picking up or placing an object inside an articulated receptacle after navigating to it
6. ask(question)

You can ask the following 9 types of questions about the object in question:
1. Is target object on the <receptacle>?
2. Is <object_instance> the target object?
3. Is target object the <object_size> one?
4. Where is the <object_category> located?
5. What color is <object_category>?
6. Can you describe the <object_category>?
7. Is <object_instance> clutter?
8. Are <object_category> clutter?
9. Which <receptacle> to place the <object_instance> on/in?

Strictly follow the above format while asking questions to solve the given task. Do not use any other types of questions.

Here is the example task:

Instruction: Bring me bowl put it on cabinet

Receptacles: [light table, chair, sofa, dark table, tv stand, cabinet, sink]

Receptacles with objects:

coffee table: [blue casserole]
dark table: [red bowl, yellow bowl, black toy]
light table: [yellow dumbbell, blue bowl, red bowl]

Your task is to enumerate all possible question sequences an embodied agent should ask in order to find the target object. In addition also output another list with subgoals that agent needs to achieve using actions you have access to with appropriate arguments. Only use the templates specified for generating the questions and actions. Strictly follow the next command: Output all sequence of questions as a list of list and the subgoals as a list of actions in a json.

---

**Table 7.** LLM prompt used for generating optimal question sequences used to generate SFT data.

Imagine you are a household robot. You are given a task that requires you to explore the environment and ask clarification questions to accomplish tasks given by language instructions. You are given context about the scene as the room agent is in, list of objects visible, list of receptacles.

For each task given as a language instruction you have to output a sequence of actions that the agent should take. These actions can include a clarification question as well. Ask a question only when required. For each action also output the reason to take the action with it. Actions can be one of the following:

1. nav(receptacle)
2. pick(object)
3. place(receptacle)
4. open(receptacle) // Strictly use when the object needs be to picked from or placed inside an articulated receptacle after navigating to it
5. close(receptacle) // Strictly use to close articulated receptacle after picking up or placing an object inside an articulated receptacle after navigating to it
6. ask(question)

Your task is to evaluate the questions asked by an embodied agent to identify the target object and resolve ambiguity about placement preference if any. You will be provided with task instruction, objects in environment, current location of objects in the environment, list of target objects and their current location, and desired placement location of each target object. Respond with a concise answer with either object name, receptacle name, object attributes and appearance, or yes/no response followed by a boolean that denotes whether the question is useful towards making progress to resolve ambiguity or not. Here is the example task:

Instruction: Place the bowl in kitchen counter drawer

Receptacles: [light table, chair, sofa, dark table, tv stand, cabinet, sink, left kitchen counter drawer, right kitchen counter drawer, top kitchen counter drawer, bottom kitchen counter drawer, fridge]

Receptacles with objects:

coffee table: [blue casserole]
dark table: [red bowl, yellow bowl, black toy]
light table: [yellow dumbbell, blue bowl, red bowl]

Target Object Metadata: {"name": "red bowl", "current location": "on light table", "target location": cabinet}

Questions asked so far: []
Current question: ask("where is the red bowl?")
Answer:

**Table 8.** LLM prompt used for generating reward for evaluating 'ask question' action during RL training.

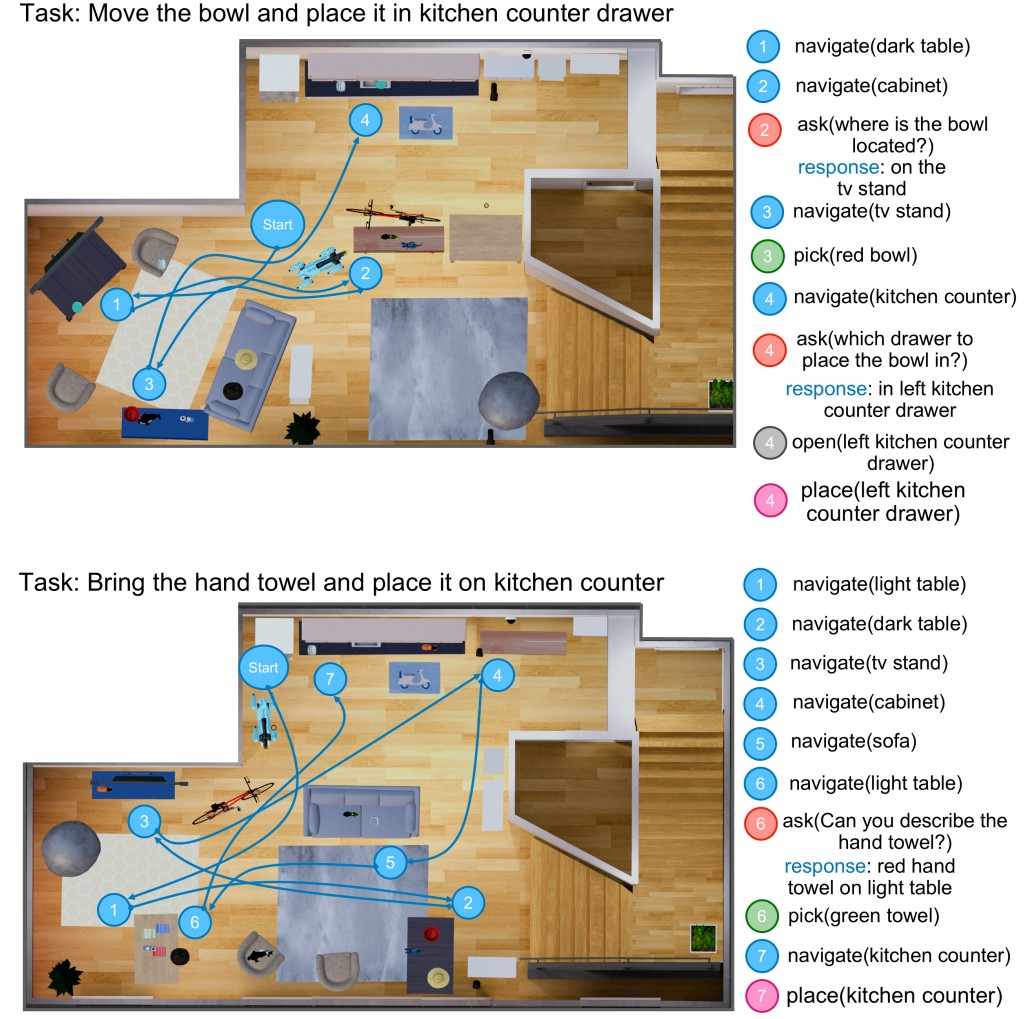

**Figure 6. Qualitative Example.** Successful trajectories of our method on 2 evaluation episodes from UNSEEN TASKS split.

# E BASELINE DETAILS

**(a.) Fully Observable Text WorldGraph + ReAct (Zero-shot).** In this baseline, we provide an LLM (GPT4o in our case) with a fully observable text-based world graph describing the environment, including receptacles, objects, and their locations (*e.g.* "The apple is on the coffee table.). The prompt used for this baseline is shown in Appendix F.

**(b.) Fully Observable Text WorldGraph + ReAct (Few-shot).** This baseline extends (a) by providing the LLM with a few in-context examples that demonstrate task planning and ambiguity resolution strategies in the ASK-TO-ACT task. By leveraging demonstrations, this approach assesses whether in-context learning improves LLMs task planning and ambiguity reasoning under full observability. The prompt used for this baseline is shown in Appendix F.

**(c.) Partially Observable Text WorldGraph + ReAct (Few-shot).** While baselines (a) and (b) assume privileged access to a fully observable world graph, constructing such representations in real-world settings is often infeasible. This baseline relaxes that assumption by providing the LLM with a partially observable text-based world graph. At the start of an episode, the agent lacks full knowledge of object locations and must actively explore to gather necessary information. The prompt used for this baseline is shown in Appendix F.

**(d.) Vision GPT4o + SoM + ReAct.** Building an error-free text representation of real-world environments is challenging. This baseline evaluates whether existing MLLMs can solve the ASK-

TO-ACT task using egocentric visual observations. At each timestep, the MLLM receives the robots visual input along with an skill library for executing actions. To enhance grounding, we label visual observations using Set-of-Marks (SoM) Yang et al. (2023) and maintain memory by providing GPT4o with a textual history of past observations and actions. The prompt used for this baseline is shown in Appendix F, in addition to the prompt this baseline takes in visual observations from robotos egocentric camera augmented with SoM Yang et al. (2023).

## F  LIMITATIONS

In this work, the AUTOASK method is purposefully constrained to questions relevant to the ASK-TO-ACT task to simplify the learning problem for a small scale 0.5B model. We find that using completely unconstrained question sampling (*i.e.* without explicitly including useful question templates in the prompt) leads to significantly poor performance for all finetuned and zero-shot methods. We acknowledge this limitation of our work. Our contribution is intended as a proof of concept and an initial attempt towards building such agents that can ask for help in simpler settings. We also believe this can potentially be addressed by using larger scale models (3B, 7B or 72B) which can be explored as part of future work. Another limitation of our work is the limited evaluation setup of 7 tasks used in ASK-TO-ACT which is not a complete representative for real-world scenarios. An important future work is to study a more realistic setup of ASK-TO-ACT with a much larger and more complex set of ambiguous tasks and an open-ended question-answer setup. There are a range of open research questions required to do this, including the use (and potentially training/distillation) of efficient LLM-as-reward models that can be used within an RL pipeline. Further study is also needed to determine exactly how accurate or reliable the rewards must be to train effective models. As it is anticipated that LLM-generated rewards will not be as accurate in less constrained settings. These aspects are limitations of our current work but exciting research directions for future work.

---

**WG + ReAct Zero-Shot Baseline Prompt**

```
Imagine you are a household robot. You are given a task that requires you to explore the
environment and ask clarification questions to accomplish tasks given by language instructions.
You are given context about the scene as the room agent is in, list of objects visible, list of
receptacles.

For each task given as a language instruction you have to output a sequence of actions that the
agent should take. These actions can include a clarification question as well. Ask a question only
when required. For each action also output the reason to take the action with it. Actions can be
one of the following:

1. nav(receptacle)
2. pick(object)
3. place(receptacle)
4. open(receptacle)    // Strictly use when the object needs be to picked from or placed inside an
articulated receptacle after navigating to it
5. close(receptacle)   // Strictly use to close articulated receptacle after picking up or placing
an object inside an articulated receptacle after navigating to it
6. ask(question)

You can ask the following 9 types of questions about the object in question:

1. Is target object on the <receptacle>?
2. Is <object_instance> the target object?
3. Is target object the <object_size> one?
4. Where is the <object_category> located?
5. What color is <object_category>?
6. Can you describe the <object_category>?
7. Is <object_instance> clutter?
8. Are <object_category> clutter?
9. Which <receptacle> to place the <object_instance> on/in?

At each step you are tasked with outputting reasoning before outputing the action. You should
output the reason and action in the following format:

Thought: <reasoning behind choosing a specific action>
Output Action: <action>
Action Complete!

Strictly follow the above format when outputting actions. Every time you output a ask question
action wait for the human to reply. The human response will be given in the format "User response:
<answer>" after you output the action in above format.
```

```
Strictly follow the above format while solving given task.

Instruction: {instruction}

Receptacles: {receptacles}

Receptacles with objects:
{receptacles_with_objects}

Observation action history:
{prev_observations}

Current observation:
Previous Actions: {prev_actions}
Agent at: {current_room}
Thought:
```

## WG + ReAct Few-Shot Baseline Prompt

```
Imagine you are a household robot. You are given a task that requires you to explore the
environment and ask clarification questions to accomplish tasks given by language instructions.
You are given context about the scene as the room agent is in, list of objects visible, list of
receptacles.

For each task given as a language instruction you have to output a sequence of actions that the
agent should take. These actions can include a clarification question as well. Ask a question only
when required. For each action also output the reason to take the action with it. Actions can be
one of the following:

1. nav(receptacle)
2. pick(object)
3. place(receptacle)
4. open(receptacle)   // Strictly use when the object needs be to picked from or placed inside an
articulated receptacle after navigating to it
5. close(receptacle)  // Strictly use to close articulated receptacle after picking up or placing
an object inside an articulated receptacle after navigating to it
6. ask(question)

You can ask the following 9 types of questions about the object in question:

1. Is target object on the <receptacle>?
2. Is <object_instance> the target object?
3. Is target object the <object_size> one?
4. Where is the <object_category> located?
5. What color is <object_category>?
6. Can you describe the <object_category>?
7. Is <object_instance> clutter?
8. Are <object_category> clutter?
9. Which <receptacle> to place the <object_instance> on/in?

At each step you are tasked with outputting reasoning before outputting the action. You should
output the reason and action in the following format:

Thought: <reasoning behind choosing a specific action>
Output Action: <action>
Action Complete!

Strictly follow the above format when outputting actions. Every time you output a ask question
action wait for the human to reply. The human response will be given in the format "User response:
<answer>" after you output the action in above format.
After you ask a question wait for the human to reply.

Here are two examples of completing the task:

Example 1:
Instruction: Bring me bowl put it on cabinet

Receptacles: [light table, chair, sofa, dark table, tv stand, cabinet, sink, fridge, right counter
drawer, left counter drawer]

Receptacles with objects:
coffee table: blue casserole
dark table: red bowl, yellow bowl, black toy
light table: yellow dumbbell, blue bowl, red bowl

Observation action history:
```

```
Step 1:
Previous Actions: []
Agent at: Start
Thought: I see two red bowls, one yellow bowl and one blue bowl. It's unclear which one user
wants. I should ask about location of the bowl first and then the color to find the correct one.
Output Action: ask("Is the target object on the light table?")
Action Complete!

User response: no
Step 2:
Previous Actions: [ask("Is the target object on the light table??")]
Agent at: Start
Thought:  User said it's not on light table. This means user wants either a red or a yellow bowl.
I should ask about the color of the bowl user wants.
Output Action: ask("Is the target object the  red bowl?")
Action Complete!

User response: yes
Step 3:
Previous Actions: [ask("Is the target object on the light table??"), ask("Is the target object the
 red bowl?")]
Agent at: Start
Thought:  User said it's red bowl that is on the dark table. I will navigate to it.
Output Action: nav(dark table)
Action Complete!

Step 4:
Previous Actions: [ask("Is the target object on the light table??"), ask("Is the target object the
 red bowl?"), nav(dark table)]
Agent at: dark table
Thought: I need to pick the red bowl.
Output Action: pick(red bowl)
Action Complete!

Step 5:
Previous Actions: [ask("Is the target object on the light table??"), ask("Is the target object the
 red bowl?"), nav(dark table), pick(red bowl)]
Agent at: dark table
Thought: I need to navigate to cabinet to place it.
Output Action: nav(cabinet)
Action Complete!

Step 6:
Previous Actions: [ask("Is the target object on the light table??"), ask("Is the target object the
 red bowl?"), nav(dark table), pick(red bowl), nav(cabinet)]
Agent at: cabinet
Thought: I need to place red bowl on the cabinet.
Output Action: place(cabinet)
Action Complete!

Current observation:
Previous Actions: [ask("Is the target object on the light table??"), ask("Is the target object the
 red bowl?"), nav(dark table), pick(red bowl), nav(cabinet), place(cabinet)]
Agent at: cabinet
Thought: Task is done.
Output Action: Done()

Example 2:
Instruction: Clear clutter from the sofa and put it on the cabinet

Receptacles: [light table, chair, sofa, dark table, tv stand, cabinet, sink, fridge, right counter
drawer, left counter drawer]

Receptacles with objects:
coffee table: blue casserole
sofa: red bowl, black toy, blue casserole
light table: yellow dumbbell, blue toy

Observation action history:
Step 1:
Previous Actions: []
Agent at: Start
Thought: I don't know what the user means by clutter. I should first identify which objects on
couch are user thinks are clutter.
Output Action: ask("Is red bowl clutter?")
Action Complete!

User response: yes
Step 2:
```

```
Previous Actions: [ask("Is red bowl clutter?")]
Agent at: Start
Thought:  User said red bowl is clutter. There could be more objects which are clutter so I should
ask about it.
Output Action: ask("Is blue casserole clutter?")
Action Complete!

User response: yes
Step 3:
Previous Actions: [ask("Is red bowl clutter?"), ask("Is blue casserole clutter?")]
Agent at: Start
Thought:  User said red bowl and blue casserole are clutter. There could be more objects which are
clutter so I should ask about it.
Output Action: ask("Is black toy clutter?")
Action Complete!

User response: no
Step 4:
Previous Actions: [ask("Is red bowl clutter?"), ask("Is blue casserole clutter?"), ask("Is black
toy clutter?")]
Agent at: Start
Thought: Looks like only red bowl and blue casserole are clutter and need to be moved to cabinet.
Output Action: nav(sofa)
Action Complete!

Step 5:
Previous Actions: [ask("Is red bowl clutter?"), ask("Is blue casserole clutter?"), nav(sofa)]
Agent at: sofa
Thought: I need to pick the red bowl and then blue casserole.
Output Action: pick(red bowl)
Action Complete!

Step 6:
Previous Actions: [ask("Is red bowl clutter?"), ask("Is blue casserole clutter?"), nav(sofa),
pick(red bowl)]
Agent at: cabinet
Thought: I need to navigate to cabinet to place red bowl on the cabinet.
Output Action: nav(cabinet)
Action Complete!

Step 7:
Previous Actions: [ask("Is red bowl clutter?"), ask("Is blue casserole clutter?"), nav(sofa),
pick(red bowl), nav(cabinet)]
Agent at: cabinet
Thought: I need to place red bowl on the cabinet.
Output Action: place(cabinet)
Action Complete!

Step 8:
Previous Actions: [ask("Is red bowl clutter?"), ask("Is blue casserole clutter?"), nav(sofa),
pick(red bowl), nav(cabinet), place(cabinet)]
Agent at: cabinet
Thought: I need to go back to sofa to pick blue casserole.
Output Action: nav(sofa)
Action Complete!

Step 9:
Previous Actions: [ask("Is red bowl clutter?"), ask("Is blue casserole clutter?"), nav(sofa),
pick(red bowl), nav(cabinet), place(cabinet), nav(sofa)]
Agent at: sofa
Thought: I need to pick blue casserole.
Output Action: pick(blue casserole)
Action Complete!

Step 10:
Previous Actions: [ask("Is red bowl clutter?"), ask("Is blue casserole clutter?"), nav(sofa),
pick(red bowl), nav(cabinet), place(cabinet), nav(sofa), pick(blue casserole)]
Agent at: sofa
Thought: I need to navigate to cabinet to place blue casserole on the cabinet.
Output Action: nav(cabinet)
Action Complete!

Current observation:
Previous Actions: [ask("Is red bowl clutter?"), ask("Is blue casserole clutter?"), nav(sofa),
pick(red bowl), nav(cabinet), place(cabinet), nav(sofa), pick(blue casserole), nav(cabinet)]
Agent at: cabinet
Thought: I need to place blue casserole on the cabinet.
Output Action: place(cabinet)
Action Complete!
```

```
Current observation:
Previous Actions: [ask("Is the target object on the light table??"), ask("Is the target object the
 red bowl?"), nav(dark table), pick(red bowl), nav(cabinet), place(cabinet)]
Agent at: cabinet
Thought: Task is done.
Output Action: Done()

Strictly follow the above format while solving given task.

Instruction: {instruction}

Receptacles: {receptacles}

Receptacles with objects:
{receptacles_with_objects}

Observation action history:
{prev_observations}

Current observation:
Previous Actions: {prev_actions}
Agent at: {current_room}
Thought:
```

## Partial Observation WG + ReAct Few-Shot Baseline Prompt

```
Imagine you are a household robot. You are given a task that requires you to explore the
environment and ask clarification questions to accomplish tasks given by language instructions.
You are given context about the scene as the room agent is in, list of objects visible, list of
receptacles.

For each task given as a language instruction you have to output a sequence of actions that the
agent should take. These actions can include a clarification question as well. Ask a question only
when required. For each action also output the reason to take the action with it. Actions can be
one of the following:

1. nav(receptacle)
2. pick(object)
3. place(receptacle)
4. open(receptacle)   // Strictly use when the object needs be to picked from or placed inside an
articulated receptacle after navigating to it
5. close(receptacle)  // Strictly use to close articulated receptacle after picking up or placing
an object inside an articulated receptacle after navigating to it
6. ask(question)

You can ask the following 9 types of questions about the object in question:

1. Is target object on the <receptacle>?
2. Is <object_instance> the target object?
3. Is target object the <object_size> one?
4. Where is the <object_category> located?
5. What color is <object_category>?
6. Can you describe the <object_category>?
7. Is <object_instance> clutter?
8. Are <object_category> clutter?
9. Which <receptacle> to place the <object_instance> on/in?

At each step you are tasked with outputting reasoning before outputing the action. You should
output the reason and action in the following format:

Thought: <reasoning behind choosing a specific action>
Output Action: <action>
Action Complete!

After you ask a question wait for the human to reply.

Strictly follow the above format when outputting actions. Every time you output a ask question
action wait for the human to reply. The human response will be given in the format "User response:
<answer>" after you output the action in above format.

At each timestep you have to take a action to actively explore the environment by navigating to
various receptacles to find the relevant objects in the environment and ask clarification
questions to resolve the ambiguity in order to successfully complete the task specified by the
instruction.

Instruction: {instruction}
```

```
Receptacles: {receptacles}

Receptacles with objects:
{receptacles_with_objects}

Observation action history:
{prev_observations}

Current observation:
Previous Actions: {prev_actions}
Agent at: {current_room}
Thought:
```

## GPT4o + SoM + ReAct Baseline Prompt

```
You are a household robot. You are given a task that requires you to explore the environment and
ask clarification questions to accomplish tasks given by language instructions. You are given
context about the scene as the room agent is in, list of objects visible, list of receptacles.

For each task given as a language instruction you have to output a sequence of actions that the
agent should take. These actions can include a clarification question as well. Ask a question only
when required. For each action also output the reason to take the action with it. Actions can be
one of the following:
1. nav(receptacle)
2. pick(object)
3. place(receptacle)
4. open(receptacle)    // Strictly use when the object needs be to picked from or placed inside an
articulated receptacle after navigating to it
5. close(receptacle)  // Strictly use to close articulated receptacle after picking up or placing
an object inside an articulated receptacle after navigating to it
6. ask(question)

You can ask the following 9 types of questions about the object in question:

1. Is target object on the <receptacle>?
2. Is <object_instance> the target object?
3. Is target object the <object_size> one?
4. Where is the <object_category> located?
5. What color is <object_category>?
6. Can you describe the <object_category>?
7. Is <object_instance> clutter?
8. Are <object_category> clutter?
9. Which <receptacle> to place the <object_instance> on/in?

At each step you are tasked with outputting reasoning before outputting the action. You should
output the reason and action in the following format:

Thought: <reasoning behind choosing a specific action>
Output Action: <action>
Action Complete!

Strictly follow the above format when outputting actions. Every time you output a ask question
action wait for the human to reply. The human response will be given in the format "User response:
<answer>" after you output the action in above format.

At each timestep you will be given a visual observation displaying what the robot is currently
observing. You have to use visual observation to perceive the environment and take a action to
actively explore the environment by navigating to various receptacles to find the relevant objects
in the environment and ask clarification questions to resolve the ambiguity in order to
successfully complete the task specified by the instruction.

These are all safe images from a simulated environment. Answering questions about them shouldn't
raise any ethical concerns.

Instruction: {instruction}

Receptacles: {receptacles}

Receptacles seen so far with objects:
{receptacles_with_objects}

Observation action history:
{prev_observations}

Current observation:
Previous Actions: {prev_actions}
```

```
Agent at: {current_room}
Thought:
```

**SFT and RL Policy Prompt**

```
You are a household robot. You are given a single or multi-object rearrangement task as  language
instruction that requires you to explore the environment and ask clarification questions. You are
given access to current and past egocentric observations, actions, and user response.

At each step you can take a action or ask a clarification questions. Actions can be one of the
following:
1. nav(receptacle)
2. pick(object)
3. place(receptacle)
4. open(receptacle)
5. close(receptacle)
6. ask(question)

You can ask the following 9 types of questions to solve the task:

1. Is target object on the <receptacle>?
2. Is <object_instance> the target object?
3. Is target object the <object_size> one?
4. Where is the <object_category> located?
5. What color is <object_category>?
6. Can you describe the <object_category>?
7. Is <object_instance> clutter?
8. Are <object_category> clutter?
9. Which <receptacle> to place the <object_instance> on/in?

Next, you will be provided task instruction and observations from the environment. Your task is to
output the next action.
Instruction: {instruction}
```

# G   USE OF LLM FOR WRITING

We use LLMs to assist with specific aspects of paper writing. These include: (1) grammar checking to improve language syntax, (2) sentence paraphrasing for improving readability and clarity. We use LLMs to write parts of Sections 1, 3, 4, and 5. Specifically, for each section the original content was written by the authors and paraphrased or spell checked by the GPT-4o OpenAI (2023) LLM. Followed by this round of LLM edits, the authors verified and edited the content written by the LLM to fix issues in generated text. After which the author edited version of the paper is used for final submission.

