# OpenReview forum: "Grounding Multimodal LLMs to Embodied Agents that Ask for Help with Reinforcement Learning"
_ICLR.cc/2026/Conference — ICLR 2026 Conference Desk Rejected Submission_

### Official Review · Reviewer_nK4L · 2025-10-17

**Soundness:** 3
**Presentation:** 3
**Contribution:** 3
**Rating:** 8
**Confidence:** 3

**Summary:**

The paper introduces ASK-TO-ACT, a benchmark and training paradigm for embodied agents that can recognize ambiguous, under-specified human instructions and ask clarifying questions before acting. To solve this challenge, the authors propose AUTOASK, a RL approach that fine-tunes multimodal large language models (MLLMs), specifically LLaVA-OneVision, into vision-language-action (VLA) policies capable of both acting and asking. Unlike imitation learning or manually engineered reward pipelines, AUTOASK employs LLM-generated dense rewards that evaluate the agent’s question quality and progress toward disambiguation. Experiments in the Habitat 3.0 simulator with 83 ReplicaCAD scenes show that AUTOASK outperforms strong zero-shot baselines (e.g., GPT-4o with ReAct) and supervised fine-tuned models by 10.4–16.5% in success rate and 8.8–20.0% in ambiguity-resolution efficiency on unseen scenes and tasks.

**Strengths:**

- ASK-TO-ACT introduces a realistic and underexplored problem, training embodied agents to ask clarification questions, bridging human-robot interaction and multimodal reasoning.

- The use of LLM-generated per-step rewards provides a scalable alternative to manual reward engineering, enabling nuanced supervision for reasoning about ambiguity.

- Results span multiple baselines (zero-shot, few-shot, SFT) and metrics (success rate, ARS, question ratio), showing consistent gains across unseen scenes and unseen tasks.

- Architecture and RL training details are presented with explicit design choices (e.g., Perceiver downsampling for long visual histories, DD-PPO optimization).

**Weaknesses:**

- Results are limited to the Habitat simulator; no real-robot validation or human-in-the-loop evaluation is presented, leaving open questions about real-world transfer.


- The LLM reward model accesses ground-truth environment state, which is unavailable in real deployments, potentially overstating feasibility.


- There is little discussion of the variance or consistency of the reward signals produced by the LLM, which is critical for RL convergence.

**Questions:**

- How would AUTOASK perform with real human responses (which can be noisy, incomplete, or contradictory) instead of simulated LLM-generated answers?

- How stable are LLM-generated rewards across different prompts or LLM variants (e.g., Llama-3 vs GPT-4)? Could reward inconsistency destabilize policy learning?

- Given reliance on privileged simulator state during training, what adaptations would be required to deploy this system on a real embodied platform like Spot or Stretch?

- What are the common failure modes, e.g., over-questioning, irrelevant clarifications, or ambiguous follow-ups—and how might the model align its questioning strategy to human tolerance?

---

> ### Author Response · Authors · 2025-11-23
> **Response to Reviewer nK4L**
>
> We thank reviewer for constructive feedback!
>
> > Q. Results are limited to the Habitat simulator; no real-robot validation or human-in-the-loop evaluation is presented, leaving open questions about real-world transfer.
>
> A. We acknowledge the complexity and necessity of real-world evals for robotics. However,  conducting real-world evaluations for such tasks at this scale and level of diversity is difficult and we would like to highlight that the focus of our work is on the challenges of training such models that require taking actions and asking natural language questions jointly rather than studying sim-to-real transfer of such methods.
>
> > Q. The LLM reward model accesses ground-truth environment state, which is unavailable in real deployments, potentially overstating feasibility.
>
> A. We would like to point out that the privileged information required by AutoAsk are “task instruction, objects in environment, current location of objects, list of target objects and their current location, and desired placement location”, information that is easily accessible for training tasks from the simulator metadata. We do acknowledge that generalizing this to different tasks would require additional metadata. However, such information is much easier to access and curate compared to manually engineering the dense reward for each subtask. We would also like to highlight that this information is required only at training time. At evaluation time the VLA policy acts autonomously only with user feedback on questions without requiring any privileged information.
>
> > Q. There is little discussion of the variance or consistency of the reward signals produced by the LLM, which is critical for RL convergence.
>
> A. Thanks for the suggestion. We conducted an analysis of the robustness of using an LLM as an answerer and the reward model for the Ask-to-Act task. Specifically, we conduct human verification using 3 LLMs as an answering module and a reward model for 100 unique task instances covering 7 task categories in Ask-to-Act. For each task, we first generate 1-3 different solutions that disambiguate target objects by asking a varying number of questions from 1-5 through heuristics. Using these solutions we first evaluate answering module accuracy by asking a human verifier to annotate whether the language responses generated by the LLM answering module are correct or not. Similarly, we also generate responses from the LLM reward models for all the question solutions and ask the human verifier to annotate whether the reward generated by the LLM reward model for the question solution is correct or not. A question solution (i.e. sequence of questions for a task) is deemed successful only if the reward model labels reward for each question successfully. Out of the 200 generated solutions for 100 tasks we ensure ~40% are failures i.e. include questions that do not disambiguate target objects successfully to ensure we are evaluating both positive and negative scenarios that occur during training. We report these results in table 1 and 2 below.
>
> | Answerer LLM            | Accuracy |
> |----------------|----------|
> | GPT-4o         | 94.5%    |
> | LLama 3.1 8B   | 85.5%    |
> | LLama 3.1 70B  | 90.2%    |
>
>
> | Reward Model LLM            | Accuracy | Precision | Recall |
> |----------------|----------|-----------|--------|
> | GPT-4o         | 77.9%    | 68.3%     | 85.4%  |
> | LLama 3.1 8B   | 66.4%    | 54.9%     | 70.0%  |
> | LLama 3.1 70B  | 71.4%    | 63.4%     | 77.7%  |
>
> > Q. How would AUTOASK perform with real human responses (which can be noisy, incomplete, or contradictory) instead of simulated LLM-generated answers?
>
> A. Due to limited time we were not able to conduct these evaluations.
>
> > Q. Given reliance on privileged simulator state during training, what adaptations would be required to deploy this system on a real embodied platform like Spot or Stretch?
>
> A. We would like to highlight that privileged information is only required by the reward model which is used only at the training time. At evaluation time the VLA policy acts autonomously only with user feedback on questions without requiring any privileged information. Therefore, when deploying the policy on real robots we do not require privileged information.
>
> > Q. What are the common failure modes, e.g., over-questioning, irrelevant clarifications, or ambiguous follow-ups—and how might the model align its questioning strategy to human tolerance?
>
> A. Thanks for the suggestion. We conducted an analysis on different failure modes and categorized them into following 4 modes:
>
> | Failure Mode                               | % Failure Episodes |
> |---------------------------------------------|--------------------|
> | Under questioning                           | 34.8%              |
> | Over-questioning                           | 65.4%              |
> | Irrelevant questions                        | 10.2%              |
> | Task planning                               | 7.7%               |

---

### Official Review · Reviewer_Tw6h · 2025-10-28

**Soundness:** 3
**Presentation:** 3
**Contribution:** 3
**Rating:** 6
**Confidence:** 3

**Summary:**

This work introduces the ASK-TO-ACT task, where an embodied agent performs single- or multi-object rearrangement in a household environment based on an under-specified instruction. Accordingly,  this paper proposes a framework that fine-tunes multimodal large language models through online reinforcement learning guided by the reward generated from a large language model. The goal is to empower agents with the abilities of asking for help with human-in-the-loop feedback. Experiments show promising results on unseen scenes or tasks.

**Strengths:**

1. The proposed task, i.e., Ask-to-Act, is interesting. It shifts the focus from purely executing instructions to recognizing ambiguity and asking for help when needed.
2. Using LLMs to generate rewords is interesting, which can reduce the dependency on expensive manual reward engineering.
3. Experiments cover unseen scenes and tasks, which is important for embodied agents that need to generalize beyond training trajectories.

**Weaknesses:**

1. Lacking real‐world evaluation.The experiments are conducted in simulated home environments . Real‐world environments are much complex.
2. While the reward design for “useful” questions is interesting, defining what counts as “useful” can be subtle—especially in real-world environments with more types of ambiguity. The method relies on privileged state information to label questions as useful, which may not hold in real settings.
3. Leveraging LLMs to generate rewards is interesting. However, it introduces concerns regarding the correctness and the robustness of these rewards—especially in large, complex environments or when the LLM provides incorrect assessments.

**Questions:**

1. How can we know the quality of LLM-generated rewards to errors in object detection or partial observations?
2. How does the proposed method compared to exploration-based strategies for deciding when to ask?
3. How does the proposed method work in real-world environments?
4. Can the agent handle multi-turn clarification?

---

> ### Author Response · Authors · 2025-11-23
> **Response to Reviewer Tw6h**
>
> We thank the reviewer for constructive feedback!
>
> > Q. Lacking real‐world evaluation.The experiments are conducted in simulated home environments . Real‐world environments are much complex.
>
> A. We acknowledge the complexity and necessity of real-world evals for robotics. However,  conducting real-world evaluations for such tasks at this scale and level of diversity is difficult and we would like to highlight that the focus of our work is on the challenges of training such models that require taking actions and asking natural language questions jointly rather than studying sim-to-real transfer of such methods.
>
> > Q. The method relies on privileged state information to label questions as useful, which may not hold in real settings.
>
> A. We would like to point out that the privileged information required by AutoAsk “task instruction, objects in environment, current location of objects, list of target objects and their current location, and desired placement location” is easily accessible for training tasks from the simulator metadata. Note we require this information only at training time which is easy to access just from the metadata of each task from the simulator. We would also like to point out that this information is basic metadata required to instantiate a task for training or evaluation in the simulator.
>
> > Q. Leveraging LLMs to generate rewards is interesting. However, it introduces concerns regarding the correctness and the robustness of these rewards—especially in large, complex environments or when the LLM provides incorrect assessments.
>
> A. Thanks for the suggestion. We conducted an analysis of the robustness of using an LLM as an answerer and the reward model for the Ask-to-Act task. Specifically, we conduct human verification using 3 LLMs as an answering module and a reward model for 100 unique task instances covering 7 task categories in Ask-to-Act. For each task, we first generate 1-3 different solutions that disambiguate target objects by asking a varying number of questions from 1-5 through heuristics. Using these solutions we first evaluate answering module accuracy by asking a human verifier to annotate whether the language responses generated by the LLM answering module are correct or not. Similarly, we also generate responses from the LLM reward models for all the question solutions and ask the human verifier to annotate whether the reward generated by the LLM reward model for the question solution is correct or not. A question solution (i.e. sequence of questions for a task) is deemed successful only if the reward model labels reward for each question successfully. Out of the 200 generated solutions for 100 tasks we ensure ~40% are failures i.e. include questions that do not disambiguate target objects successfully to ensure we are evaluating both positive and negative scenarios that occur during training. We report these results in table 1 and 2 below.
>
> | Answerer LLM            |  Accuracy |
> |----------------|----------|
> | GPT-4o         | 94.5%    |
> | LLama 3.1 8B   | 85.5%    |
> | LLama 3.1 70B  | 90.2%    |
>
> | Reward Model LLM            | Accuracy | Precision | Recall |
> |----------------|----------|-----------|--------|
> | GPT-4o         | 77.9%    | 68.3%     | 85.4%  |
> | LLama 3.1 8B   | 66.4%    | 54.9%     | 70.0%  |
> | LLama 3.1 70B  | 71.4%    | 63.4%     | 77.7%  |
>
> > Q. How can we know the quality of LLM-generated rewards to errors in object detection or partial observations?
>
> A. We would like to highlight that LLM-generated rewards use privileged information about environment state directly from the simulator. Therefore, there are no errors in state information due to object detectors or partial observations. All errors can be attributed to LLM’s task-specific reasoning ability and underspecification of prompt.
>
> > Q. How does the proposed method compare to exploration-based strategies for deciding when to ask?
>
> A. We are not aware of any prior works that use exploration based strategies to decide when and what to ask for help that are directly applicable to Ask-to-Act tasks. Especially, since Ask-to-Act requires an agent to not only know when to ask for help but also what to ask makes it much more challenging for prior works.
>
>
> > Q. Can the agent handle multi-turn clarification?
>
> A. Yes, it does! We specifically created tasks as part of our Ask-to-Act task that require holding a multi-turn dialogue. Specifically, the Clean Clutter, Preference Reasoning, and Compositional Ambiguity Reasoning tasks require an agent to ask anywhere from 2-6 questions to resolve ambiguity as demonstrated by examples in Fig. 1 and qualitative example of policy behavior shown in Fig. 4 and Fig. 6 of the main paper.
>
> We are happy to answer any follow-ups or concerns reviewer has. If they think our response has addressed their concerns we'd appreciate if they could acknowledge it and adjust the rating for our paper.

---

### Official Review · Reviewer_NUjL · 2025-10-31

**Soundness:** 2
**Presentation:** 3
**Contribution:** 3
**Rating:** 4
**Confidence:** 4

**Summary:**

This paper introduces ASK-TO-ACT, a task where embodied agents must resolve ambiguity in under-specified instructions by asking clarification questions. The authors propose AUTOASK, which fine-tunes multimodal large language models (MLLMs) as vision-language-action (VLA) policies using online reinforcement learning with LLM-generated rewards. The method is evaluated in Habitat 3.0 on tasks requiring agents to fetch and place objects while disambiguating user intent through natural language questions. AUTOASK outperforms zero-shot baselines and supervised fine-tuning approaches by 10.4-16.5% on unseen scenes and tasks.

**Strengths:**

- The paper demonstrates that subgoal rewards alone are insufficient for learning effective question-asking behavior. This motivates the need for dense, question-specific rewards.
- The reward ablation study (Table 2) and analysis of question budgets provide valuable insights into the trade-offs between task success and user interaction burden.
- AUTOASK outperforms both zero-shot GPT-4o baselines and behavior cloning supervised fine-tuning, demonstrating the effectiveness of RL training for this task.
- The ASK-TO-ACT benchmark includes diverse ambiguity types (attribute recognition, spatial reasoning, object size, compositional ambiguity) that test different reasoning capabilities.

**Weaknesses:**

- The authors claim their LLM-based reward generation is more scalable than manually designed rewards, yet their method requires extensive privileged information including "task instruction, objects in environment, current location of objects, list of target objects and their current location, and desired placement location." This level of privileged access is arguably no more scalable than hand-crafted rewards and would not generalize to real-world settings where such information is unavailable.
- The agent is limited to 9 predefined question templates, which fundamentally undermines the claim of natural language interaction. The authors acknowledge unconstrained question generation performs poorly but dismiss this limitation by suggesting larger models might help without empirical validation.
- While the paper shows performance differences between policies trained with different rewards (e.g., subgoal rewards), it provides insufficient analysis of what specific behavioral changes lead to these improvements.
- The heavy reliance on privileged environment information for reward generation suggests the method would not transfer to other embodied AI tasks without significant manual engineering per domain.

**Questions:**

- Can the authors provide results using the LLM reward model without privileged information to substantiate scalability claims? How would the method perform if the LLM reward model only had access to the same partial observations as the agent, making it truly scalable to real-world deployment?
- What specific behavioral patterns emerge in agents trained with subgoal rewards versus those with question-specific rewards? Understanding these differences would strengthen the paper's contributions.
- Have the authors tested larger MLLMs (>7B parameters) with unconstrained question generation as suggested? This is critical for validating whether the approach can scale to more natural interactions.

---

> ### Author Response · Authors · 2025-11-23
> **Response to Reviewer NUjL**
>
> We thank the reviewer for constructive feedback!
>
> > Q. The level of privileged access AutoAsk requires is arguably no more scalable than hand-crafted rewards and would not generalize to real-world settings where such information is unavailable.
>
> > Q. The heavy reliance on privileged environment information for reward generation suggests the method would not transfer to other embodied AI tasks without significant manual engineering per domain.
>
> A. We do not share this view. We would like to point out that the privileged information required by AutoAsk “task instruction, objects in environment, current location of objects, list of target objects and their current location, and desired placement location” is easily accessible for training tasks from the simulator metadata. Note we require this information **only at training time** which is easy to access just from the metadata of each task from the simulator. We would also like to point out that this information is basic metadata required to instantiate a task for training or evaluation in the simulator. Additionally, training VLA policies in the real-world with RL is especially difficult even for simple tasks like object manipulation which is easy to verify. Instantiating tasks which require human input for real-world RL is an open research problem that is out of scope of this work.
>
> We do acknowledge that generalizing this approach to generate rewards to different tasks would require additional metadata and prompt engineering. However, such information is much easier to access and curate compared to manually engineering the dense reward for each subtask.
>
> > Q. The agent is limited to 9 predefined question templates, which fundamentally undermines the claim of natural language interaction. The authors acknowledge unconstrained question generation performs poorly but dismiss this limitation by suggesting larger models might help without empirical validation.
>
> A. We acknowledge the limitation of using 9 predefined question templates as part of our policy prompt to enable training of small-scale VLA policies. However, due to limited compute we can not validate training larger scale models in an unconstrained setting as part of this work. We have revised the limitations section to acknowledge this limitation.
>
> > Q. While the paper shows performance differences between policies trained with different rewards (e.g., subgoal rewards), it provides insufficient analysis of what specific behavioral changes lead to these improvements.
>
> A. Thanks for the suggestion. We would like to highlight the key reason AutoAsk reward performs the best is it provides dense supervision for questions asked by the agent based on whether it reveals any new information than previously asked questions or not. Specifically, the LLM only gives a positive reward for a question if it helps make progress towards identifying the target object. Such dense supervision is missing from subgoal and sparse success rewards as these rewards only evaluate task success and intermediate rearrangement subgoals and lack the ability to provide dense supervision.
>
> > Q. Can the authors provide results using the LLM reward model without privileged information to substantiate scalability claims?
>
> A. We tested this baseline and it leads to reward model achieving near 0 performance as the key information about the task i.e. the target object metadata is missing. In Ask-to-Act the key piece information required by an LLM to evaluate questions asked by an agent is task instruction, current object states, target object, desired target locations. Removing any of this information makes an LLM incapable of evaluating the questions asked by an agent, therefore, leading to incorrect rewards. We’d also like to highlight that the privileged information is needed only at training time which is easy to access just from the metadata of each task from the simulator.
>
> > Q. What specific behavioral patterns emerge in agents trained with subgoal rewards versus those with question-specific rewards?
>
> A. We conducted an analysis on different behavior of models trained with subgoal reward vs LLM generated reward and find that policies trained with subgoal reward demonstrate low frequency for asking questions i.e. often ask too few questions than required. In addition, they ask irrelevant questions and do not learn the deductive reasoning to ask useful questions that will help disambiguate the target object.
>
> > Q. Have the authors tested larger MLLMs (>7B parameters) with unconstrained question generation as suggested?
>
> A. No, unfortunately due to limited compute we could not train policies using larger MLLMs >= 7B parameters.
>
> We are happy to answer any follow-ups or concerns reviewer has. If they think our response has addressed their concerns we'd appreciate if they could acknowledge it and adjust the rating for our paper.

---

> > ### Comment · Reviewer_NUjL · 2025-11-25
> > **Thank you and follow up**
> >
> > Thank you for the detailed response. I greatly appreciate the clarifications.
> >
> > I had a follow up below regarding the llm reward scalability.
> >
> > Given the metadata and list of questions, do the authors have an estimate of the complexity of a manually designed reward using a rule-based criteria, and how much effort the llm judge prompt engineering saves? E.g., if the object is not in view, reward for questions like “Where is the <object category> located?”, etc. Are there cases where such a decision criteria could not account for some edge cases in the author’s current training setting? This would help better understand the tradeoffs of manual reward versus the author’s llm judge.

---

> > > ### Author Response · Authors · 2025-11-25
> > > **Response to follow up**
> > >
> > > We believe hand-crafting rewards for the general problem of evaluating natural language questions in any context using rule-based logic is infeasible and not scalable. Consider, the example shown in Fig. 1 of the paper, the agent can ask a diverse range of questions “Where is the cup located?”, “Where is the target cup currently placed at?”, “What is the color of the cup?”, “Is the cup located on a light table?”, etc. To be able to evaluate and answer these questions we would need a robust language parsing logic that can map questions to relevant target object metadata available in simulation which is intractable.
> > >
> > > In the limited context of tasks set up as part of the Ask-to-Act benchmark it might be possible to write hand-crafted evaluation criteria for only a subset of all possible questions using language parsers, heuristics, and simulation metadata. But this would require extremely complex logic to handle a variety of edge cases. It would require a significant amount of effort to build such heuristics and handle all edge cases. In contrast, we propose to use an LLM to evaluate and answer questions which can be arbitrarily scaled to any paraphrase or type of question without additional heuristics. Once an effective prompt with adequate environment and target object metadata is designed, it generalizes without requiring task-specific engineering. Moreover, these prompts are general criteria or rubrics that can be reused across different types of tasks with different kinds of ambiguity and do not require task-specific heuristics in contrast to hand-crafted rules.

---

### Official Review · Reviewer_w8Mf · 2025-11-01

**Soundness:** 3
**Presentation:** 2
**Contribution:** 2
**Rating:** 4
**Confidence:** 3

**Summary:**

The papers proposes to use RL to fun-tune the VLA to VLA by LLM-generated rewards. Also, the paper focus on enable the RL tuned VLA to do multi-round ask-reasoning and act by ultilizing online RL.

**Strengths:**

1. The paper is easy to read and follow.

2. The problem this paper focus, that under the case that the user give a unclear instruction or goal, the agent need to be able to do ask and reasoning then act in multiple round, is very practical questions in the community, even lots of the current fundation robot model only focus on train simple policies under assumptions that both the environment an dinstruction is determinstic and clear.

**Weaknesses:**

1. The problem this paper focus is practical and meaningful, but potential concerns existed in LLM-generated rewards: Although ASK-TO-ACT aims to model human-robot clarification dialogues, the “user” responses are simulated by an LLM using privileged state. This avoids actual human ambiguity, noise, or miscommunication. Thus, the realism of the claimed human-robot interaction remains limited.

2. The paper only has experiemt on simple sumulated env, but not real env, which for this specific problem, real env is the most emportant: All baselines are either handcrafted LLM agents (ReAct) or simple SFT on synthetic trajectories. In order to support the conclusion, the author might need to consider some baseline like human-feedback RL (RLAIF / RLHF) approaches, fundation models VLAs such as OpenVLA or Pi0, reward-learning baselines like Text2Reward or Eureka-style code-generated rewards applied directly.

**Questions:**

See the above weakness. Besides those, I have further concern that:
1. WIth the existed experiment design, I feel there still lacks some analysis of why RL with LLM-generated rewards helps. Section 5.3 shows the reward choice will affect the result, but overall lack interpretation of all the design and results.

---

> ### Author Response · Authors · 2025-11-23
> **Response to the Reviewer w8Mf**
>
> We thank the reviewer for constructive feedback!
>
> > Q. The problem this paper focuses on is practical and meaningful, but potential concerns exist in LLM-generated rewards: Although ASK-TO-ACT aims to model human-robot clarification dialogues, the “user” responses are simulated by an LLM using privileged state. This avoids actual human ambiguity, noise, or miscommunication. Thus, the realism of the claimed human-robot interaction remains limited.
>
> A. We acknowledge that simulating human ambiguity using an LLM might not be an exact representation of noise or miscommunication in real world scenarios. However, these LLMs are imperfect which naturally leads to some noise and miscommunication in answers generated by the LLM. We present analysis in section 5.4 of revised paper of answer module accuracy and present a few examples of miscommunication and noise these agents exhibit. We also present the LLM answering module accuracy metrics computed using a human evaluation study for easier access as part of this response.
>
> | LLM            | Accuracy |
> |----------------|----------|
> | GPT-4o         | 94.5%    |
> | LLama 3.1 8B   | 85.5%    |
> | LLama 3.1 70B  | 90.2%    |
>
>
> > Q. The paper only has experiment on simple simulated env, but not real env, which for this specific problem, real env is the most important:
>
> A. Thanks for the feedback. We would like to highlight that the focus of our work is on the challenges of training agents in such scenarios rather than studying sim-to-real transfer of such methods. Our framework allows for fast iteration on methods, after which the best methods can be tested in the real world. Conducting real-world evaluations for such tasks at this scale and level of diversity is quite challenging and its own line of research that can be done for future work.
>
> > Q. Missing baselines like OpenVLA, Pi, Eureka, TextToReward, and RLHF.
>
> A. Thanks for the suggestion. We wanted to clarify that baselines such as OpenVLA, Eureka, and Pi are not directly applicable to the Ask-to-Act task as these methods are trained for low-level control and the focus of our work is to train a high-level VLA planner that can interleave task execution with asking questions. Additionally, the VLA architecture we use is similar to these methods with differences in output action space and training algorithm.
>
>
> > Q. WIth the existing experiment design, I feel there still lacks some analysis of why RL with LLM-generated rewards helps. Section 5.3 shows the reward choice will affect the result, but overall lacks interpretation of all the design and results.
>
> A. Thanks for the suggestion. We would like to highlight the key reason AutoAsk reward performs the best is it provides dense supervision for questions asked by the agent based on whether it reveals any new information than previously asked questions or not. Specifically, the LLM only gives a positive reward for a question if it helps make progress towards identifying the target object. Such dense supervision is missing from subgoal and sparse success rewards as these rewards only evaluate task success and intermediate rearrangement subgoals and lack the ability to provide dense supervision.
>
>
> We are happy to answer any follow-ups or concerns reviewer has. If they think our response has addressed their concerns we'd appreciate if they could acknowledge it and adjust the rating for our paper.

---

### Note · Program_Chairs · 2026-01-05
**Submission Desk Rejected by Program Chairs**

The author's updated paper includes author names, violating double-blind, and consequently the submission must be desk rejected.